# A multi-protein receptor-ligand complex underlies combinatorial dendrite guidance choices in *C. elegans*

Wei Zou[1], Ao Shen[2], Xintong Dong[1], Madina Tugizova[1], Yang K Xiang[2], Kang Shen[1]*

[1]Department of Biology, Howard Hughes Medical Institute, Stanford University, Stanford, United States; [2]Department of Pharmacology, University of California, Davis, Davis, United States

**Abstract** Ligand receptor interactions instruct axon guidance during development. How dendrites are guided to specific targets is less understood. The *C. elegans* PVD sensory neuron innervates muscle-skin interface with its elaborate dendritic branches. Here, we found that LECT-2, the ortholog of leukocyte cell-derived chemotaxin-2 (LECT2), is secreted from the muscles and required for muscle innervation by PVD. Mosaic analyses showed that LECT-2 acted locally to guide the growth of terminal branches. Ectopic expression of LECT-2 from seam cells is sufficient to redirect the PVD dendrites onto seam cells. LECT-2 functions in a multi-protein receptor-ligand complex that also contains two transmembrane ligands on the skin, SAX-7/L1CAM and MNR-1, and the neuronal transmembrane receptor DMA-1. LECT-2 greatly enhances the binding between SAX-7, MNR-1 and DMA-1. The activation of DMA-1 strictly requires all three ligands, which establishes a combinatorial code to precisely target and pattern dendritic arbors.

*For correspondence: kangshen@stanford.edu

## Introduction

Neural circuit assembly requires precise guidance of both axons and dendrites. Several families of secreted proteins, such as netrin/UNC-6 and Slit, have been extensively studied for their functions as axon guidance cues (*Brose et al., 1999*; *Colamarino and Tessier-Lavigne, 1995*; *Hedgecock et al., 1990*; *Kennedy et al., 1994*; *Kidd et al., 1999*; *Li et al., 1999*; *Serafini et al., 1994*). Region-specific expression of these cues establishes gradients or borders that guide axonal navigation through receptor molecules on the growth cone. To date, most of the receptor-ligand interactions involve one ligand and its cognate receptor or receptor-coreceptor pair (*Dickson, 2002*; *Huber et al., 2003*; *Yu and Bargmann, 2001*).

Dendritic growth and branching is often cell-type specific and follows a stereotyped trajectory, suggestive of precise dendrite guidance mechanisms (*Dong et al., 2015*; *Jan and Jan, 2010*). However, the molecular nature of these mechanisms remains to be fully understood. Studies in model organisms have begun to shed light on what guides dendrites. Dendritic targeting of *Drosophila* motoneurons is regulated by midline signaling molecules, including Slit-Robo and Netrin-Fra (*Brierley et al., 2009*; *Mauss et al., 2009*). A recent study showed that homotypic interactions between the Dscam2 and Dscam4 genes are required for the precise targeting of the L4 dendrites in the *Drosophila* visual system (*Tadros et al., 2016*). For fly olfactory projection neurons, dendritic targeting is regulated by cell-autonomous function of graded expression of Sema-1a (*Komiyama et al., 2007*). Interaction between integrin on the dendrite and laminin on the basement membrane restricts the two dimensional structure of dendritic arbors of *Drosophila* class IV dendritic arborization neurons and ensures dendritic self-avoidance (*Han et al., 2012*; *Kim et al., 2012*).

Notably, there is considerable overlap between axon and dendrite guidance cues, as the above-mentioned molecules are also well characterized axon guidance cues. Still, dendrite morphogenesis is distinct from axon morphogenesis in several ways—dendrites often exhibit specific branching patterns, can self-avoid and form specialized sensory dendrite-peripheral tissue interactions (*Corty et al., 2009*; *Dong et al., 2015*; *Grueber and Sagasti, 2010*) — so it is likely that there are distinct mechanisms for dendrite morphogenesis. Indeed, the highly unusual *Drosophila* DSCAM gene generates numerous alternatively spliced variants that mediate self-avoidance of dendrites through stringent isoform-specific homophilic interactions (*Hughes et al., 2007*; *Matthews et al., 2007*; *Soba et al., 2007*). Similarly, accumulating evidence suggests that the mammalian protocadherin genes can also create a large number of splice variants and mediate self-avoidance (*Lefebvre et al., 2012*; *Wu and Maniatis, 1999*). In order to uncover additional dendrite-specific guidance mechanisms, we use the *C. elegans* PVD neuron to study cell-cell interactions that precisely pattern complex dendrites.

The two *C. elegans* PVD neurons develop highly-branched yet stereotyped dendritic arbors, one on each side of the worm (*Albeg et al., 2011*). They first develop anterior-posteriorly oriented primary (1°) dendrites. During the larval 2 (L2) stage to L3 stage, 2° dendrites emerge and mostly grow along the ventral-dorsal axis. When 2° dendrites reach the border of the outer body wall muscles, they start to form T-shaped 3° dendrites. Finally, during the L3 to L4 stage, 4° dendrites branch out from the 3° dendrites and grow along the ventral-dorsal axis. All the dendrites grow along the surface of the skin, or epidermis. Importantly, the numerous 4° dendrites are sandwiched between the epidermis and body wall muscles (*Figure 1A and B*) (*Albeg et al., 2011*). Functional studies suggest that one of PVD's functions is to regulate body posture similar to a vertebrate proprioceptor (*Albeg et al., 2011*). Therefore, the stringent muscle innervation by the PVD dendrites is likely to have functional importance. Previous studies showed that SAX-7/L1CAM and MNR-1/Menorin form a co-ligand complex on the epidermis to regulate the patterning of PVD dendrites (*Dong et al., 2013*; *Salzberg et al., 2013*). On the PVD dendritic membrane, the leucine-rich repeat protein DMA-1 acts as the cognate receptor that physically interacts with SAX-7 and MNR-1 (*Dong et al., 2013*; *Liu and Shen, 2012*; *Salzberg et al., 2013*). SAX-7 forms stripes on the epidermis where 3° and 4° dendrites develop (*Figure 1B* and *Figure 1—figure supplement 1*) (*Dong et al., 2013*; *Liang et al., 2015*; *Salzberg et al., 2013*).

In this study, we identified LECT-2 as a novel dendritic guidance cue that guides the growth of PVD dendrites. Loss of *lect-2* severely affects the dendritic patterning in the PVD neurons. LECT-2 is mainly secreted from the muscles, and acts as both a short-range cue to guide the growth of 4° dendrites and a long-range cue to direct the growth of 2° and 3° dendrites. Seam cell expressed LECT-2 causes mistargeting of PVD dendrites onto seam cells. We also found that LECT-2 directly interacts with SAX-7, and it dramatically increases the binding efficiency between SAX-7, MNR-1 and DMA-1. Together, LECT-2, SAX-7, MNR-1 and DMA-1 form a multiprotein ligand-receptor complex. Because the activation of DMA-1 strictly requires the presence of all three ligands, the PVD dendrites receives a combinatorial code from the environment to precisely guide their morphogenesis.

## Results

### *lect-2* is required for dendritic patterning of the PVD neurons

To visualize the morphology of PVD dendritic arbors, we utilized a membrane targeted GFP marker driven by a PVD neuron specific promoter (*ser2prom3>myr-gfp*) (*Figure 1C*) (*Tsalik et al., 2003*). We also labeled body wall muscles and seam cells using cytosolic mCherry expressed under tissue-specific promoters (*Pmyo-3>mcherry* and *Pnhr-81>mcherry*, respectively) as they are part of the growth environment of the PVD neurons (*Figure 1A–E*). In wild-type animals, most of the 2° dendrites grow perpendicularly from the 1° dendrites and form 'T-shaped' 3° dendrites. PVD also elaborates orderly 4° dendrites that are sandwiched between the epidermis and the body wall muscles (*Figure 1A–E and I–K*). To identify novel factors important for PVD dendrite morphogenesis, we carried out an unbiased forward genetic screen and identified two mutants, *wy935* and *wy953*, in which the patterning of PVD dendrites were severely affected in 100% of animals. Compared with wild type PVD, these mutants contained more numerous and disorganized 2° dendrites, which completely failed to form 'T' shaped 3° dendrites. In addition, no 4° dendrites formed in these mutants

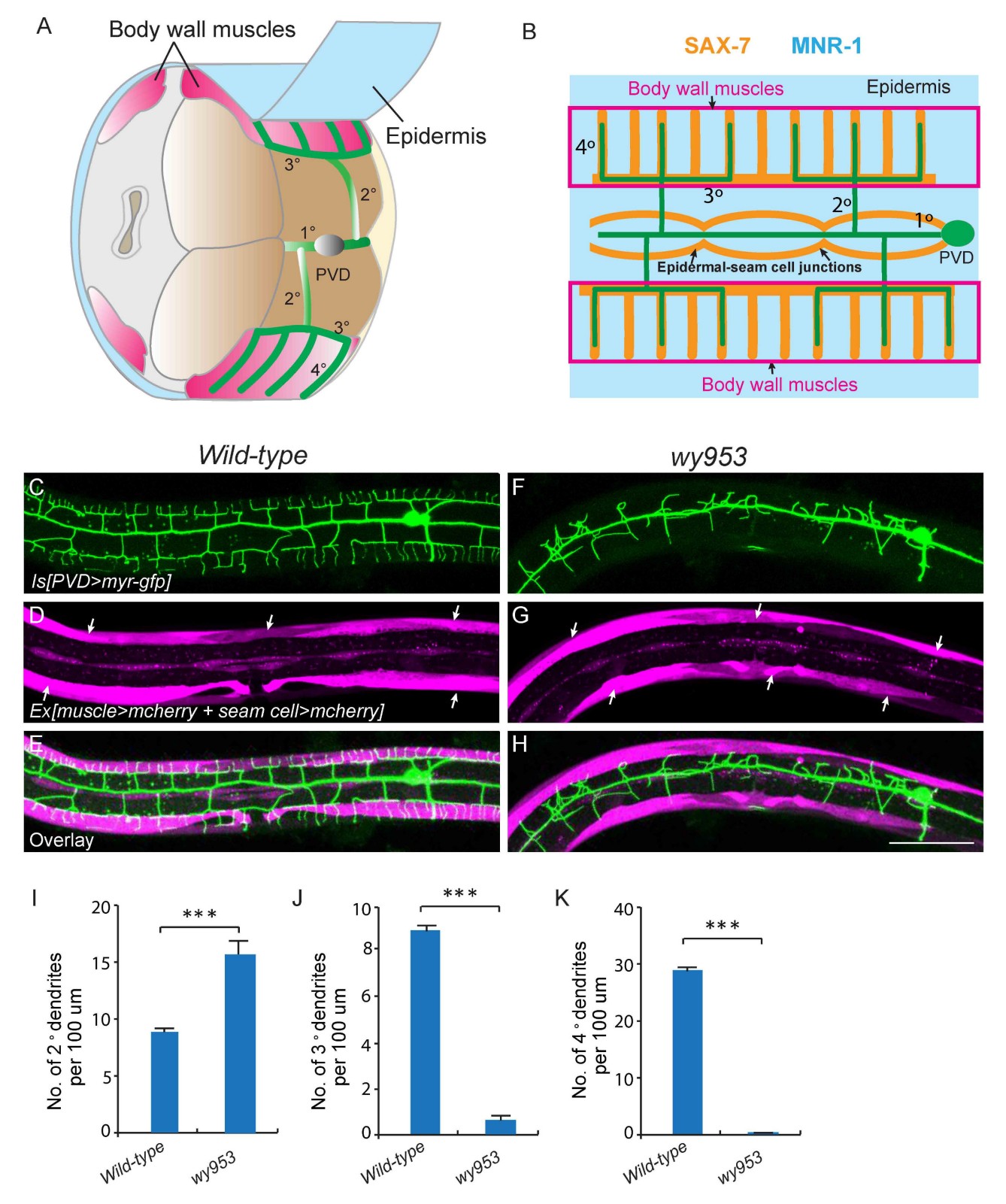

**Figure 1.** *wy953* mutants show severe dendritic guidance defects. (**A**) A cartoon showing the morphology of part of a PVD dendritic tree and its growth environment (adapted from ***Albeg et al., 2011***). 1°: primary dendrites; 2°: secondary dendrites; 3°: tertiary dendrites; 4°: quaternary dendrites; Note that tertiary dendrites grow along the border of outer body wall muscles, and quaternary dendrites are sandwiched between epidermis and body wall muscles. (**B**) A cartoon showing the morphology of PVD dendritic arbors and the localization of SAX-7 and MNR-1, which regulate PVD dendritic

*Figure 1 continued on next page*

*Figure 1 continued*

patterning. MNR-1 is evenly distributed on the epidermis. SAX-7 is enriched along tertiary lines, quaternary stripes and epidermal-seam cell junctions. Red boxes highlight the quaternary dendrites that are sandwiched between the epidermis and body wall muscles. (C–H) Confocal images showing PVD dendrites (labeled in green using transgene *ser2prom3>myr-gfp* here and in subsequent figures), body wall muscles (arrows, labeled in magenta using transgene *Pmyo-3>mcherry* here and in subsequent figures) and seam cells (labeled in magenta using transgene *Pnhr-81>mcherry* here and in subsequent figures, close to the 1° dendrites of PVD neurons) in wild-type (C–E) and *wy953*mutant (F–H) worms. Scale bar: 50 μm. (I–K) Quantifications showing the number of 2°, 3° and 4° dendrites in a 100 μm region anterior to the cell body of PVD neurons. Student's t-test was used for statistical analysis. ***p<0.0001. Data are represented as mean ± SEM. 16 animals were quantified for each genotype.

The following figure supplement is available for figure 1:

**Figure supplement 1.** PVD dendrites do not grow along all the SAX-7 positive loci.

(*Figure 1F–K*). These two alleles failed to complement each other, suggesting that they affected the same gene. Using single nucleotide polymorphism (SNP) mapping and transgene rescue experiments, we identified the causative point mutations in a novel gene *K05F1.5*, which we named *lect-2* (worm homolog of the human leukocyte cell-derived chemotaxin-2). Both *wy935* and *wy953* contained G to A mutations, resulting in glycine to glutamic acid mutations at amino acids 259 and 103 in the LECT-2 protein, respectively (*Figure 2A*). When placed over *maDf4* (a deficiency allele that covers the entire *lect-2* genomic locus), *wy935* showed a similar PVD dendritic morphogenesis defect to that of *wy935* homozygous animals, suggesting that *wy935* is likely to be a null allele. We also analyzed *ok2617*, a deletion allele of *lect-2*, in which a large portion of intron #2, the entirety of exon #3 and intron #3, and a small part of exon #4 are deleted (*Figure 2A*). We found that *ok2617* caused an indistinguishable PVD dendritic morphogenesis defect as that of *wy935* and *wy953*, suggesting that all three alleles are likely to be nulls (*Figure 2B*). The defects were fully rescued by a single copy transgene in which *lect-2* genomic DNA was driven under its endogenous promoter (*Figure 2C–F*).

*lect-2* encodes a conserved protein homologous to human leukocyte cell-derived chemotaxin-2 (LECT2) (*Figure 2—figure supplement 1*). Similar to human LECT2, worm LECT-2 contains a predicted signal peptide at its N-terminus (1–20 amino acids), and two tandem M23 superfamily peptidase domains that are analogous to each other (*Figure 2A* and *Figure 2—figure supplement 1*). Interestingly, based on the alignment of the two M23 domains, *wy953* and *wy935* represent the exact same mutations on the homologous region in the N- or C- terminal M23 domains, respectively (*Figure 2—figure supplement 1*).

## *lect-2* functions in the same genetic pathway as *sax-7*, *mnr-1* and *dma-1*

The PVD dendritic morphogenesis defects in *lect-2* mutants were indistinguishable from that of *sax-7* and *mnr-1* mutants. *sax-7* and *mnr-1* encode a co-ligand complex on the surface of the epidermis, which is recognized by the PVD dendritic receptor DMA-1. Previous studies have revealed that *sax-7*, *mnr-1* and *dma-1* function in the same genetic pathway to regulate PVD dendritic morphogenesis (*Dong et al., 2013*; *Liu and Shen, 2012*; *Salzberg et al., 2013*). Similar to the *lect-2* mutants, deficiencies in *sax-7*, *mnr-1* or *dma-1* also severely affected the patterning of 2° dendrites and formation of 3° dendrites and 4° dendrites (*Figure 3B–E, G and I–K*). We sought to determine whether *lect-2* functions in the *sax-7/mnr-1/dma-1* genetic pathway. Double mutants between *lect-2* and *sax-7/mnr-1* were indistinguishable from any of the single mutants (*Figure 3B–F and I–K*). *dma-1* mutants showed more severe dendritic morphogenesis defects than that of *lect-2*, which was not further enhanced in the *dma-1; lect-2* double mutants (*Figure 3B,G–K*). Together, our genetic analyses suggest that *lect-2* functions in the same genetic pathway as *sax-7*, *mnr-1* and *dma-1*.

## LECT-2 is mainly secreted from muscles and localizes to neuropils and commissures

To determine the site of action of *lect-2*, we created a single copy transgene (*wyTi3*) in which both *lect-2* genomic DNA and *mcherry* were driven under the *lect-2* promoter and separated by a spliced leader SL2 (*Frokjaer-Jensen et al., 2014*). Two loxP sites were introduced to flank the entire transgene (*loxp-Plect-2>lect-2:: SL2::mcherry-loxp*). *wyTi3* fully rescued the PVD dendritic morphogenesis

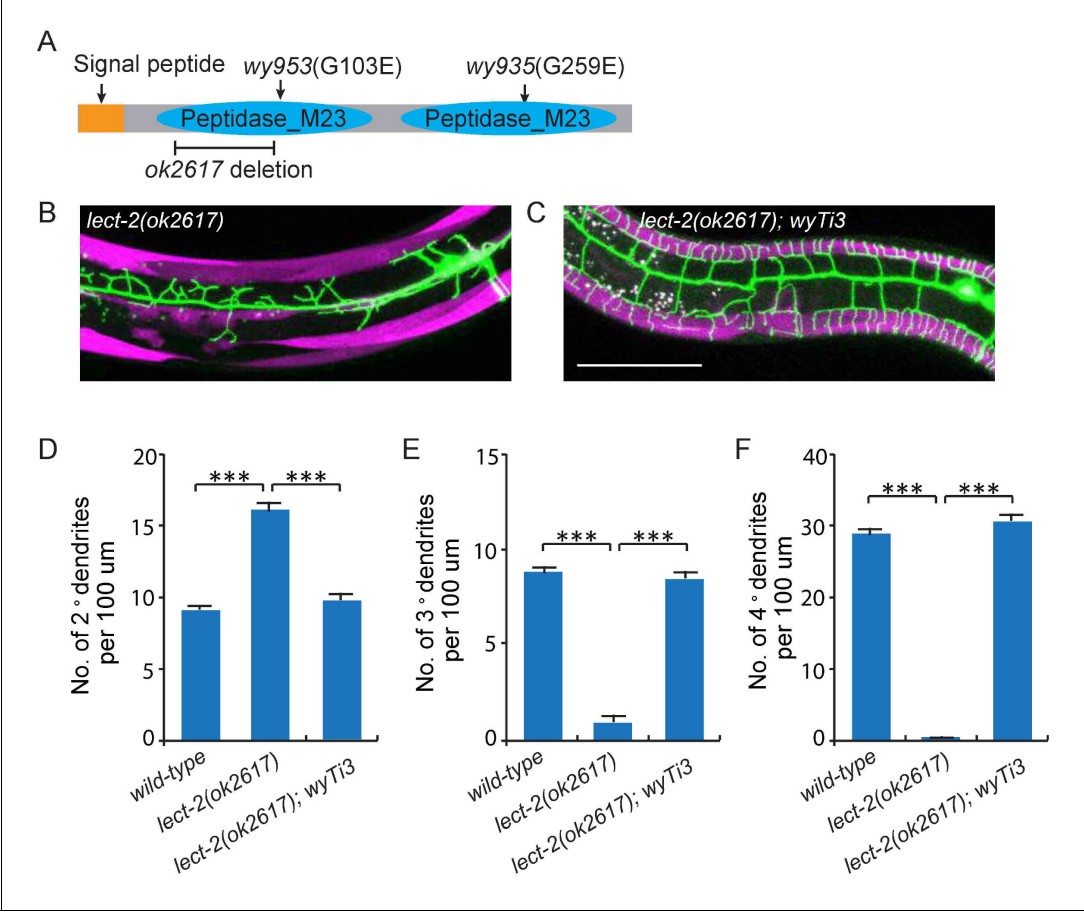

**Figure 2.** *Cloning and rescue of lect-2.* (**A**) Schematic of the LECT-2 protein with alleles indicated. (**B–C**) Confocal images showing the PVD dendritic arbors in *lect-2(ok2617)* mutants and *lect-2(ok2617); wyTi3* [*loxp -Plect-2>lect-2::SL2:: mcherry-loxp*]. Body wall muscles were labeled in both images, while seam cells were labeled in **B** but not **C**. Scale bar: 50 μm. (**D–F**) Quantifications showing the number of 2°, 3° and 4° dendrites in a 100 μm region anterior to the cell body of PVD neurons. One way ANOVA with the Dunnett's correction was used for statistical analysis. ***p<0.0001. Data are represented as mean ± SEM. 16, 12 and 12 animals were quantified for each genotype, respectively.

The following figure supplement is available for figure 2:

**Figure supplement 1.** LECT-2 is homologous to human LECT2.

defects of *lect-2*(*ok2617*) worms (*Figure 2C–F*). Unlike the ubiquitous expression of *sax-7*, or the epidermis-specific expression of *mnr-1*, expression of *lect-2* was mainly found in the body wall muscles at the L3/L4 stage, when PVD develops higher-order dendrites (*Figure 4A*) (*Dong et al., 2013*; *Salzberg et al., 2013*). In addition, several head neurons and ventral nerve cord neurons also express *lect-2* because they are labeled by the transcriptional reporter (*Figure 4A* and *Figure 4— figure supplement 1*).

To determine the localization of endogenous LECT-2 protein, we generated a *yfp* knock-in strain by CRISPR/Cas9-mediated homologous recombination (*Dickinson et al., 2013*). Consistent with the N-terminal signal peptide prediction, we observed bright YFP signals in coelomocytes, the macrophage-like cells that take up proteins secreted by other tissues into the body cavity (*Figure 4B*) (*Fares and Greenwald, 2001*). Notably, no signal from the transcriptional reporter was detected in coelomocytes (*Figure 4A*). Importantly, in addition to the bright fluorescence in the coelomocytes, YFP was also detected as stripe patterns that correspond to neuropils and commissures. In particular, LECT-2::YFP localizes to the sublateral nerve cord and muscle borders where the 3° PVD dendrite elaborates and commissures where many of the 2° PVD dendrites grow along (*Figure 4B*). This

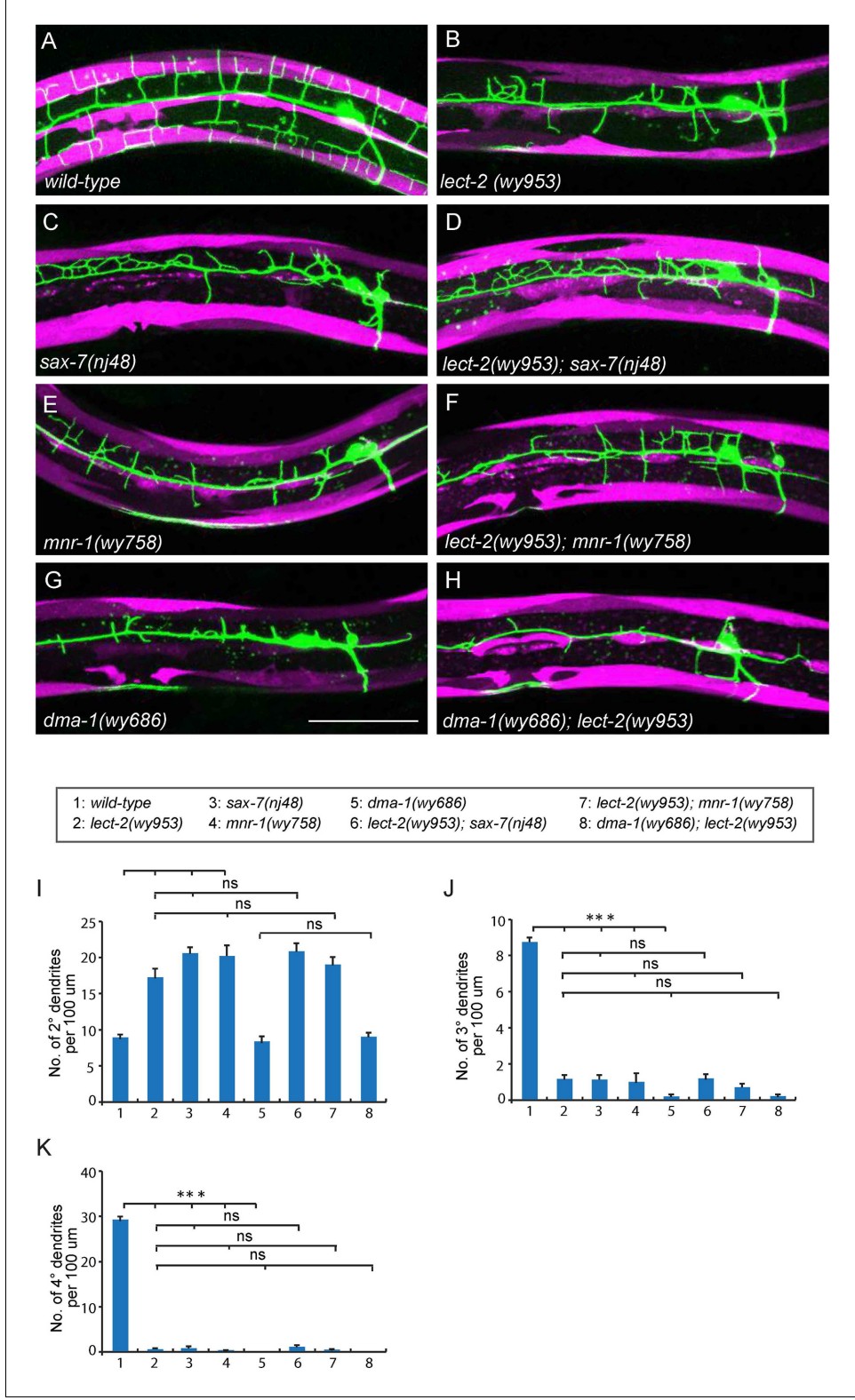

**Figure 3.** *lect-2* functions in the same genetic pathway as *sax-7*, *mnr-1* and *dma-1*. (A–H) Confocal images showing PVD dendritic arbors, body wall muscles and seam cells in (A) wild-type; (B) *lect-2(wy953)*; (C) *sax-7(nj48)*; (D) *lect-2(wy953); sax-7(nj48)*; (E) *mnr-1(wy758)*; (F) *lect-2(wy953); mnr-1(wy758)*; (G) *dma-1(wy686)* and (H) *dma-1 (wy686); lect-2(wy953)* mutants. Scale bar: 50 μm. (I–K) Quantifications of (I) number of 2° dendrites; (J) number of 3° dendrites; (K) number of 4° dendrites in a 100 μm region anterior to the cell body of PVD neurons. One way

*Figure 3 continued on next page*

*Figure 3 continued*

ANOVA with the Tukey correction was used for statistical analysis. ***p<0.0001. ns: not significant. Data are represented as mean ± SEM. 16 animals were quantified for each genotype, except for *lect-2; sax-7* (13 animals were quantified).

*yfp* insertion in the endogenous *lect-2* locus did not affect the PVD morphology except a subtle increase in the number of terminal branches, suggesting that the staining largely represents the endogenous protein localization pattern (*Figure 4C–F*).

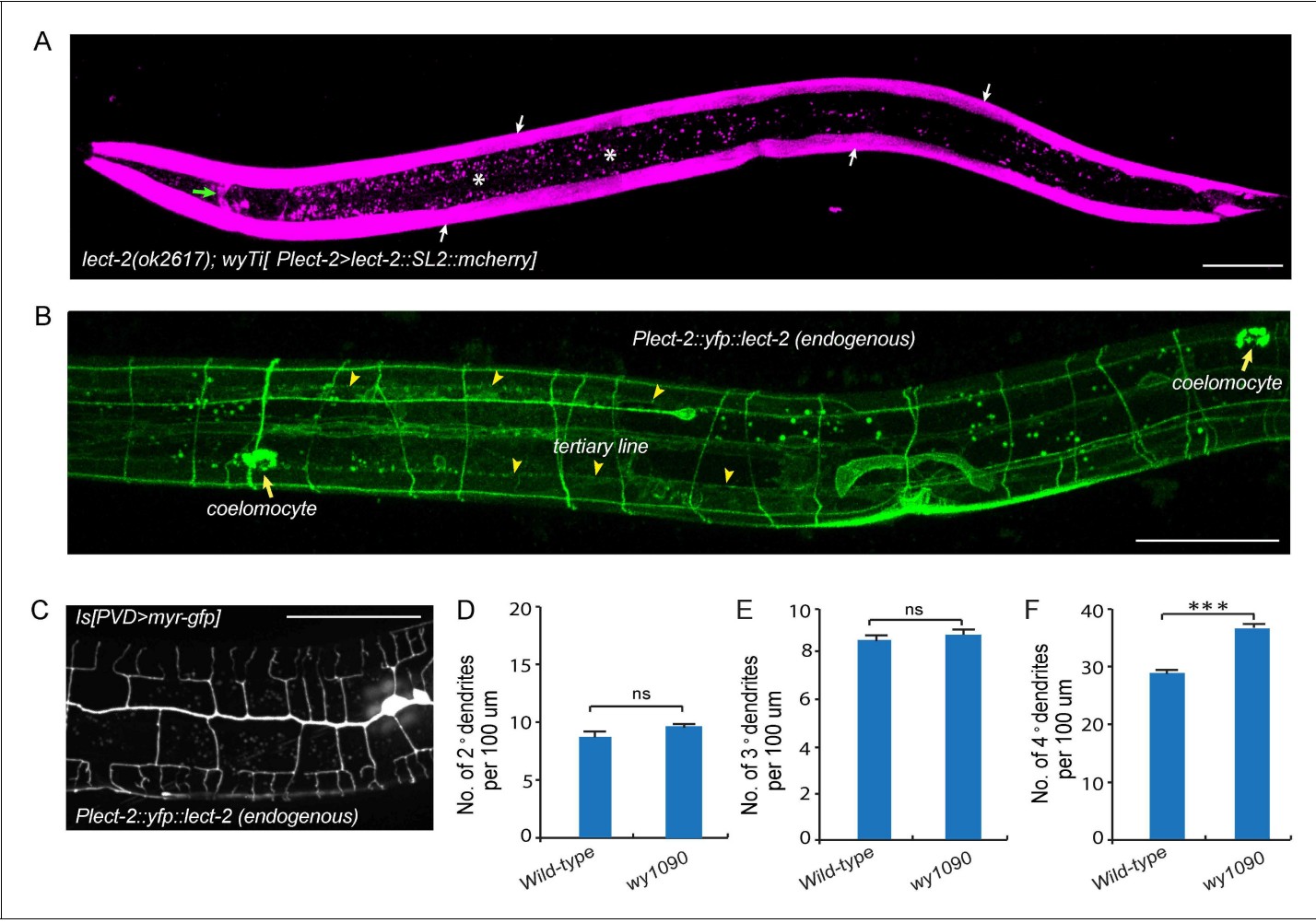

**Figure 4.** Transcriptional and translational expression patterns of *lect-2*. (A) A confocal image showing the transcriptional expression pattern of *lect-2* in an L4 stage animal. White arrows: body wall muscles. Green arrow: head neurons. Asterisks: auto fluorescence from the gut granules in the intestinal cells. Scale bar: 50 μm. (B) A confocal image showing the translational expression pattern of endogenously expressed YFP::LECT-2 in an L4 stage animal. Scale bar: 50 μm. (C) A confocal image showing the morphology of PVD dendritic arbors of a *yfp::lect-2* knock-in animal at L4 stage. Arrow heads: tertiary line along the border of outer body wall muscles. Arrows: coelomocytes. Scale bar: 50 μm. (D–F) Quantifications showing the number of 2°, 3° and 4° dendrites in a 100 μm region anterior to the cell body of PVD neurons. Student's t-test was used for statistical analysis. ***p<0.0001. ns: not significant. Data are represented as mean ± SEM. 16 animals were quantified for each genotype.

The following figure supplement is available for figure 4:

**Figure supplement 1.** *lect-2* is expressed in some ventral cord neurons.

## LECT-2 acts as a short-range cue to guide 4° dendrites to innervate muscles and a long-range cue to guide 2° dendrites and 3° dendrites

To understand where LECT-2 is required for its function, we performed three mosaic experiments to pinpoint the tissue specificity and mode of actions for this protein. In the first experiment, we expressed body wall muscle-specific Cre in the *lect-2(ok2617); wyTi3 [loxp-Plect-2>lect-2::SL2:: mcherry-loxp]* strain to eliminate muscle expression of LECT-2 (*Ruijtenberg and van den Heuvel, 2015*). When muscle expression was eliminated as shown by the lack of mCherry signal in the muscle cells, we found that the formation of 3° dendrites were largely unaffected (*Figure 5B and D*). While the number of 2° dendrites was still higher than the wild type controls, the 2° dendrites in these animals were much more regular in their growth direction and they also gave rise to 'T' shaped 3° dendrites (*Figure 5B–C*). In contract, the growth of 4° dendrites onto muscles was severely affected in these animals (*Figure 5B and E*). In two independent muscle-specific Cre strains, the number of 4° dendrites innervating muscles was decreased by 62.1% and 74.8%, respectively (*Figure 5E*). Thus, LECT-2 from muscles is not absolutely required to pattern 2° dendrites and 3° dendrites, but it is required for 4° dendrite formation.

The *C. elegans* body wall musculature consists of 95 rhomboid shaped, unfused muscle cells that are arranged into four rows of staggered cells along the longitudinal axis. Each muscle cell is innervated by 4° dendrites from a single or a few adjacent menorahs (the dendritic arbor unit that consists of all the 4° branches from the same 3° dendrite). To further understand how muscle secreted LECT-2 promotes 4° dendrites growth, we created another mosaic transgenic strain in which only a single muscle or a few muscles produced functional LECT-2 in the *lect-2(ok2617)* mutant background. The LECT-2 -expressing muscles were also labeled by mCherry. Interestingly, we found that the transgene was able to robustly restore the innervation of 4° dendrites selectively onto *lect-2*(+) muscles (*Figure 6A–C and F*). On average, 11.4 dendrites were found to innervate a single *lect-2*(+) muscle (a single muscle in L4/young adult worms is roughly 100 μm) (*Figure 6F*). In striking contrast, few terminal dendrites were found growing on the *lect-2*(-) muscles (1.7 dendrites per non-transgenic muscle) (*Figure 6A–C and F*). On the contrary, the growth direction of 2° dendrites and the formation of 'T' shaped 3° dendrites were significantly rescued in both *lect-2(+)* and *lect-2(-)* zones (*Figure 6A–E*). These results argue strongly that muscle derived LECT-2 functions as a short ranged guidance cue to ensure that each muscle is innervated by 4° dendrites. As the mean time, it functions as a long-range cue to pattern the 2° and 3° dendrites. We suspect that LECT-2 might be easily diffusible in the body cavity where 2° and 3° dendrites reside, but much less diffusible in the narrow space between the epidermis and body wall muscles where 4° dendrites form. Consistent with this idea, when we over-expressed LECT-2 in *lect-2* mutants using high-copy transgenes and strong promoters from several tissues, including the PVD neurons, epidermis, body wall muscles and pharyngeal muscles, not only the patterning of 2° dendrites and 3° dendrites but also that of 4° dendrites were robustly rescued in each case (*Figure 6—figure supplement 1*). These results suggest that overexpression of LECT-2 can overcome the putative diffusion barrier into the muscle-epidermis interspace and promote the 4° dendrite formation.

## LECT-2 can function as an instructive cue to guide dendrite growth

The overexpressed LECT-2 from several tissues rescued the dendrite patterning defects, arguing that LECT-2 plays a permissive function in PVD dendrite morphogenesis, whose spatial pattern and source of secretion is not essential. However, with low level of expression in the muscle cells which is more likely to mimic physiological conditions, LECT-2 appears to act locally as an instructive cue to guide 4° dendrites to innervate each muscle. To further test if *lect-2* can function as an instructive cue to guide dendrite pattern, we sought to create an artificial expression pattern of LECT-2 and ask if the PVD dendrite follow the LECT-2 expression pattern. In the *lect-2* mutant background, we expressed LECT-2 in the seam cells, a row of epidermal cells located roughly along the primary dendrites of PVD neurons. Interestingly, *seam cell>lect-2* did not rescue the branch formation defect of 4° dendrites. Instead, it caused dendrites to grow around the seam cells (*Figure 7A*). Quantitatively, we observed that about 36.1% of 2° dendrites (n = 929) grew along the epidermal-seam cell junctions in *lect-2(ok2617)* mutants carrying a *seam cell>lect-2* transgene, whereas only 1.8% (n = 540) and 3.0% (n = 518) of 2° dendrites normally grow along the epidermal-seam cell junctions in wild-type animals and *lect-2(ok2617)* mutants carrying a *muscle>lect-2* transgene, respectively

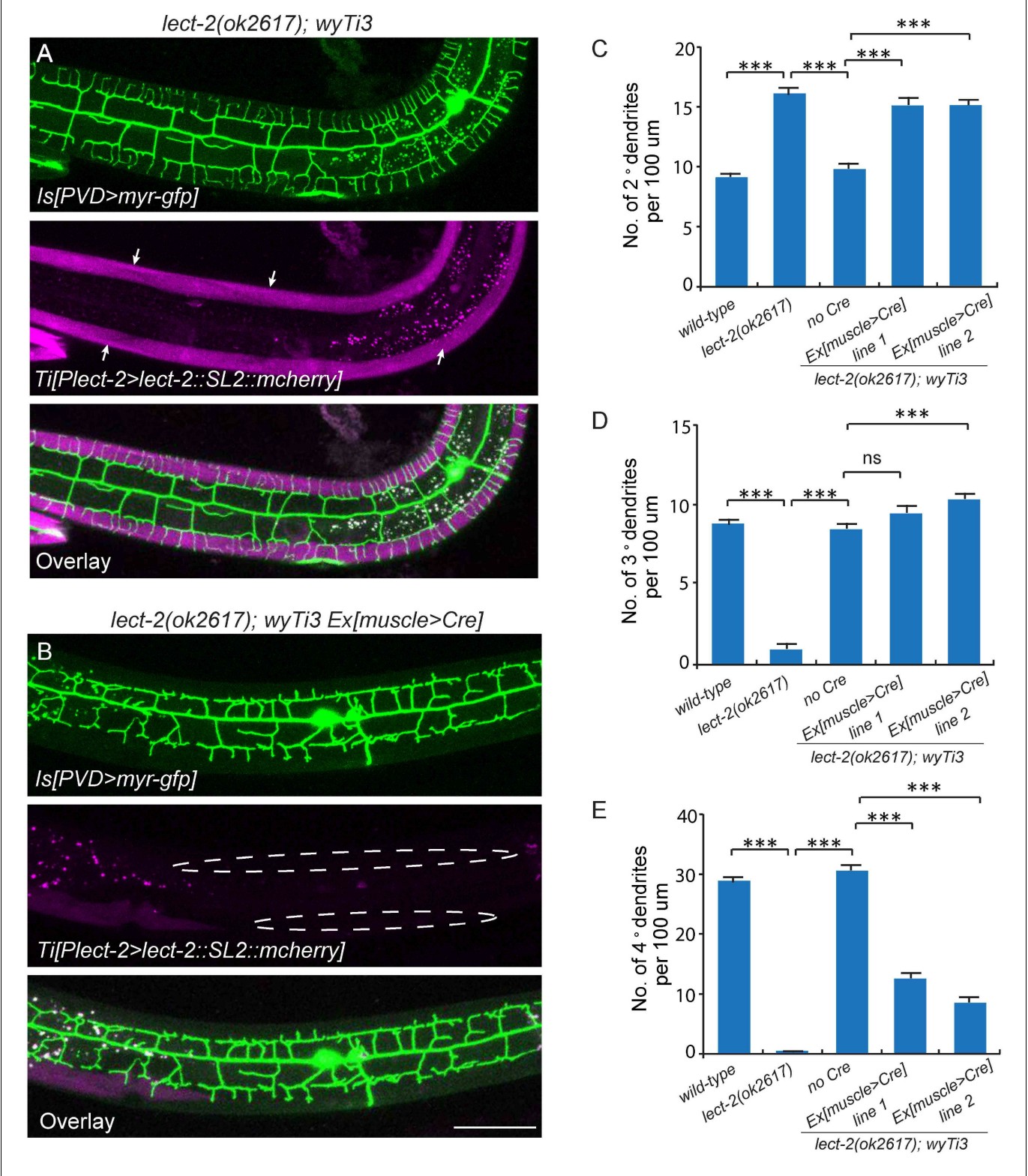

**Figure 5.** Body wall muscle-expressed LECT-2 is required for the patterning of 4° but not 2° or 3° dendrites. (A–B) Confocal images showing the PVD dendritic arbors (labeled in green using transgene *ser2prom3>myr-gfp*) in *lect-2(ok2617); wyTi3 [loxp-Plect-2>lect-2::SL2:: mcherry-loxp]* without (**A**) or with (**B**) muscle expressed Cre transgenes. Note that all the body wall muscles were labeled by mCherry in **A**, while most of the muscles (labeled by dashed lines) were not labeled in **B**. Scale bar: 50 µm. (**C–E**) Quantifications of number of 2°, 3° and 4° dendrites in a 100 µm region anterior to the cell

*Figure 5 continued on next page*

*Figure 5 continued*

body of PVD neurons. One way ANOVA with the Tukey correction was used for statistical analysis. ***p<0.0001. Data are represented as mean ± SEM. 16 animals were quantified for each genotype, except for *lect-2(ok2617)* (12 animals were quantified).

(*Figure 7A–B and F*). In addition, seam cell expressed LECT-2 also partially rescued the targeting and formation of 2°dendrites and 3° dendrites (*Figure 7A* and *Figure 6—figure supplement 1*).

As *lect-2* functions in the same genetic pathway as *sax-7*, *mnr-1* and *dma-1*, and normal PVD dendritic guidance requires all four genes, we asked whether the ectopic seam cell targeting caused by seam cell LECT-2 expression also required the other three genes. Indeed, mutations in *sax-7*, *mnr-1* or *dma-1* largely abolished the seam cell targeting of PVD dendrites (*Figure 7C–F*). Thus the dendritic guidance function of LECT-2 requires SAX-7, MNR-1 and DMA-1. Together, these mosaic experiments demonstrate that muscle expressed LECT-2 plays a local, instructive role to form 4° dendrite formation. In the meantime, it also plays a long-range, permissive role for 2° and 3° dendrite patterning.

## LECT-2 physically interacts with SAX-7

Previous studies showed that SAX-7 forms specific patterns on the epidermis to instruct PVD dendritic morphogenesis of the 3° and 4° dendrites (*Dong et al., 2013*; *Liang et al., 2015*; *Salzberg et al., 2013*). When expressed alone in a wild-type background, YFP::LECT-2 was detected along the 3° dendrites as well as many neuropils and commissures, which was reminiscent of the pattern of SAX-7 (*Figure 4B*). When we co-expressed LECT-2::GFP and SAX-7::mcherry under their endogenous promoters, we found the GFP and mCherry formed co-localized stripes where 3° and 4° dendrites grew along, suggesting that LECT-2 is recruited to SAX-7 stripes (*Figure 8A–C*). Supporting this idea, the SAX-7-like staining of YFP::LECT-2 was completely abolished in *sax-7* but not *mnr-1* mutants, while the coelomocyte staining was not affected in either of mutant (*Figure 8D–G*). In addition to the normal coelomocyte staining, the transcription of *lect-2* was not obviously affected in *sax-7* mutants, suggesting that the loss of SAX-7-like staining is unlikely due to defects in the transcription, translation or secretion of LECT-2 (*Figure 8—figure supplement 1*). These results suggest that SAX-7 but not MNR-1 is required to localize LECT-2 through a potentially direct link. To test if SAX-7 and LECT-2 physically interact with each other, we expressed GFP-tagged SAX-7 and HA-tagged LECT-2 in *Drosophila* S2 cells and performed co-immunoprecipitation (co-IP) experiments. The SAX-7::GFP was detected both as full length protein (SAX-7 FL) and a truncated protein product (SAX-7 CTF1). This is consistent with existing literature that SAX-7 is cleaved by a protease in its third FNIII domain (*Hadwiger et al., 2010*; *Kiefel et al., 2012*; *Pocock et al., 2008*; *Salzberg et al., 2013*; *Sasakura et al., 2005*; *Wang et al., 2005*). Both SAX-7 FL and SAX-7 CTF1 robustly co-precipitated with LECT-2::HA (*Figure 8H–I*). To further narrow down the binding region, we performed co-IP experiments using the SAX-7 extracellular domains (ECD). We generated both the naturally existing truncated SAX-7 ECD containing the last two FnIII domains, as well as a full length ECD by mutating the cleavage site. We observed robust co-precipitation of LECT-2 with both truncated protein and the full length ECD (*Figure 8H,J–K*, and *Figure 8—figure supplement 2*). Thus, the last two FnIII domains of SAX-7 ECD are likely sufficient to bind to LECT-2.

## LECT-2 forms a multi-protein complex with SAX-7, MNR-1 and DMA-1

Since our genetic analyses showed that *lect-2* functioned in the same pathway as *sax-7*, *mnr-1* and *dma-1*, we sought to determine whether LECT-2, SAX-7, MNR-1 and DMA-1 form a multi-protein ligand-receptor complex. We performed single-molecule pull-down (SiMPull) analysis, which is an imaging based, quantitative immunoprecipitation assay (*Figure 9A*) (*Jain et al., 2011*; *Jain et al., 2012*). We generated a *C. elegans* strain in which the endogenously expressed DMA-1 was tagged by a FLAG tag, and also carried a transgene to express GFP-tagged LECT-2 and mCherry-tagged SAX-7 under their respective endogenous promoters (*Dong et al., 2016*). DMA-1::FLAG was pulled down using a biotinylated anti-FLAG antibody from whole worm lysates. The pull-down fraction was then subjected to fluorescence microscopy, where LECT-2::GFP and SAX-7::mCherry were directly visualized through their fused fluorescent tags.

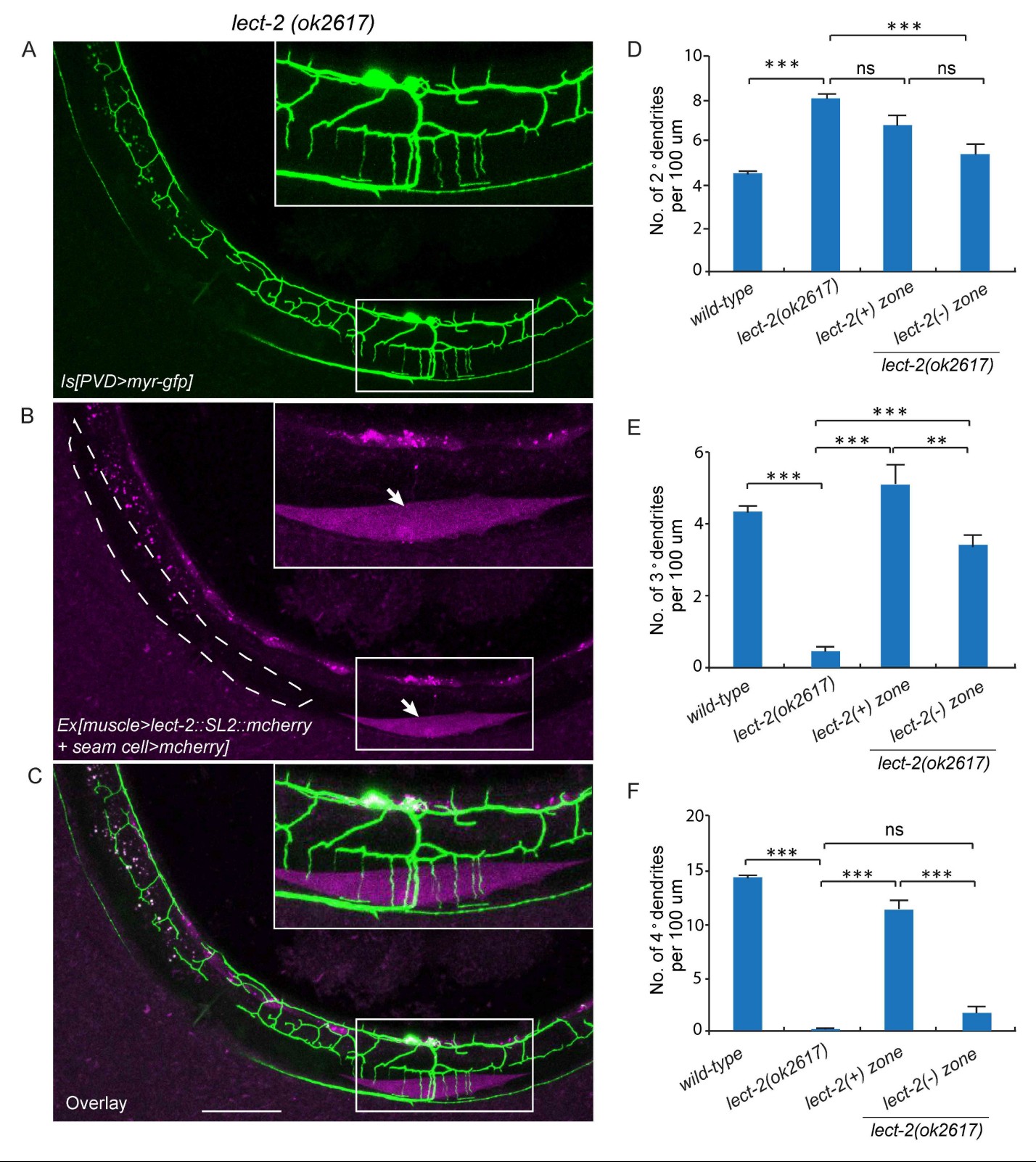

**Figure 6.** Muscle mosaic analysis. (A–C) Confocal images showing PVD dendritic arbors, body wall muscles (arrow, labeled in magenta using transgene *Phlh-1>lect-2::SL2::mcherry*) and seam cells in a *lect-2(ok2617)* mutant animal which carried muscle-expressed LECT-2 transgene. The inset images are enlarged views (two fold) of the regions indicated by the boxes. Note that these images are partially lateral views and only the ventral half of PVD dendritic arbors could be imaged. Scale bar: 50 μm. (D) Quantification of number of 2°, 3° and 4° dendrites in *lect-2(+)* zone and *lect-2(-)* zone.. For this

*Figure 6 continued on next page*

*Figure 6 continued*

quantification, number of dendrites growing in different zones was counted separately. The length of muscles along the anterior-posterior axis was also quantified. Dendritic density = number of dendrites/ length of muscles. Number of dendrites per 100 μm in *lect-2(+)* zone and *lect-2(-)* zone were shown. For each mosaic animal, either the dorsal or the ventral half of the PVD dendritic arbors was quantified, as only partially dorsal-up or ventral-up, but not fully lateral-up worms could be identified as muscle mosaic worms and used for imaging and quantification. Similar way was used to quantified *wild-type* and *lect-2(ok2617)* animals. One way ANOVA with the Tukey correction was used for statistical analysis. \*\*\*p<0.0001. Data are represented as mean ± SEM. 11 animals were quantified for each group.

The following figure supplement is available for figure 6:

**Figure supplement 1.** PVD patterning defect of *lect-2* mutants is rescued by over-expressed LECT-2 secreted from multiple types of tissues.

Three strains of worm lysates were compared using this assay. In the test strain, all three transgenes were present in untagged wild-type *mnr-1* background. In one control strain, *dma-1::flag* was replaced with untagged endogenous wild-type *dma-1* locus, while the rest of the genotypes were identical to those of the test strain. In the second control strain, all three transgenes were present in a *mnr-1* null mutant background. As expected, the expression level of LECT-2::GFP and SAX-7::mCherry were comparable among the three samples (*Figure 9—figure supplement 1*). In the control sample without the *dma-1::flag*, no fluorescence signals above the background were detected, demonstrating the specificity of this assay (*Figure 9B–D and K*). In the test sample, we detected colocalized LECT-2::GFP and SAX-7::mCherry fluorescence spots, indicating that LECT-2, SAX-7 and DMA-1 likely exist in the same protein complex. The number of fluorescent spots in mCherry and GFP channels was similar, suggestive of a roughly one-to-one association between SAX-7 and LECT-2 (*Figure 9E–G and K*). About 55% of LECT-2::GFP and SAX-7::mCherry signals co-localized with each other (*Figure 9E–G and L*). The incomplete colocalization in the assay arises mainly from inactive fluorophores. At single molecule level, protein fluorophores such as GFP are subject to misfolding or incomplete maturation (*Ulbrich and Isacoff, 2007*). It was estimated that about 75% GFP molecules are fluorescently active (*Jain et al., 2011*). Assuming this ratio, after correction of inactive fluorophores, theoretically, more than 92% LECT-2 and SAX-7 would co-localized at single molecule level. Notably, both the SAX-7 and LECT-2 signals were lost in the *mnr-1* mutant background suggesting the complex formation is dependent on endogenous MNR-1 (*Figure 9H–K*). Together with the previous results in the SAX-7/DMA-1/MNR-1 complex (*Dong et al., 2013*; *Salzberg et al., 2013*), our results suggest that LECT-2, SAX-7, MNR-1 and DMA-1 likely exist as a single protein complex.

## LECT-2 dramatically increases the binding efficiency between SAX-7, MNR-1 and DMA-1

Previous studies showed that SAX-7, MNR-1 and DMA-1 form a complex to promote dendritic stabilization and branching (*Dong et al., 2013*; *Salzberg et al., 2013*). To study how LECT-2 acts together with SAX-7, MNR-1 and DMA-1 to guide PVD dendritic growth, we sought to determine whether LECT-2 facilitates the formation of the receptor-ligand complex by increasing the binding efficiency between SAX-7, MNR-1 and DMA-1. First, we utilized the *Drosophila* S2 cell aggregation assay to test this idea. Consistent with our previous observation, DMA-1::RFP-expressing cells aggregated with cells co-expressing SAX-7::GFP and MNR-1::GFP after mixing and incubation in conditioned medium collected from untransfected control cells for more than 40 min, but not at earlier time points (*Figure 10—figure supplement 1*) (*Dong et al., 2013*). However, when these two groups of cells were incubated in LECT-2-containing S2 cell medium, we observed robust cell aggregate formation after only 10 min, suggesting that LECT-2 likely increased the binding efficiency between DMA-1, SAX-7 and MNR-1 (*Figure 10—figure supplement 1*).

To test this idea more directly, we expressed MNR-1::GFP, DMA-1::RFP, SAX-7::HA and LECT-2::FLAG in *Drosophila* S2 cells and detected protein interactions at single molecule level with SiMPull experiments. After cell lysis, SAX-7::HA was pulled down using biotinylated anti-HA antibody. Co-expressed DMA-1::RFP and MNR-1::GFP were directly visualized through their fused fluorescent tags. Consistent with our previous results, we found that MNR-1::GFP and DMA-1::RFP were pull-downed by SAX-7::HA without adding LECT-2::FLAG. The interactions were specific as the very low

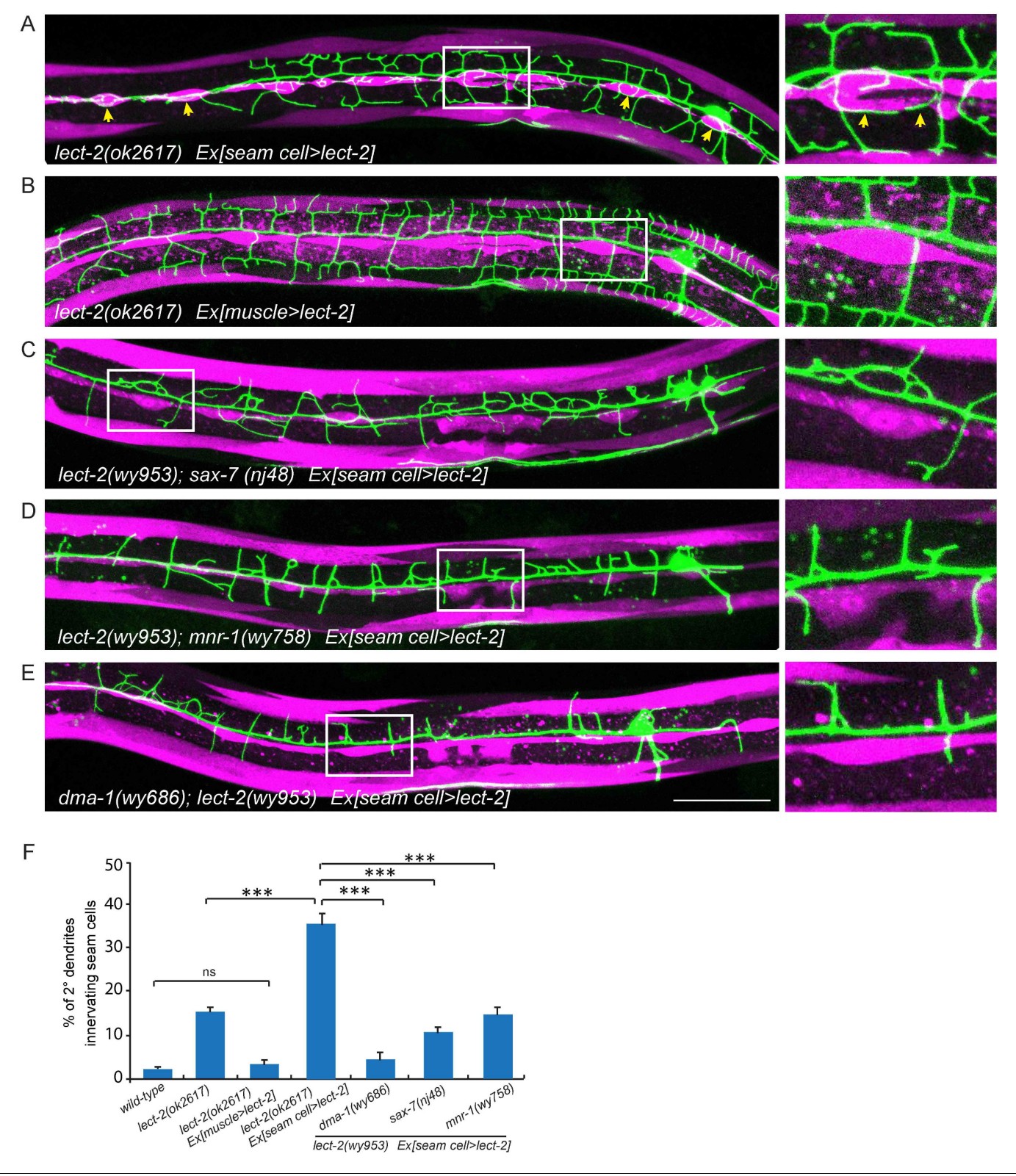

**Figure 7.** Seam cell-secreted LECT-2 could misguide dendrites onto seam cells. (E) Confocal images showing PVD dendritic arbors, body wall muscles (labeled in magenta using transgene *Pmyo-3>mcherry* (for **A**, **C–E**) or *Phlh-1> lect-2::SL2::mcherry* (for **C**)) and seam cells. The inset images are enlarged views (2 fold) of the regions indicated by the boxes. Arrows: dendrites innervating seam cells. Scale bar: 50 μm. (**F**) Quantifications of percentage of 2°

*Figure 7 continued on next page*

Figure 7 continued

dendrites innervating seam cells. One way ANOVA with the Tukey correction was used for statistical analysis. ***p<0.0001. ns: not significant. Data are represented as mean ± SEM. Number of 2° dendrites quantified for each genotype: 540, 489, 518, 929, 179, 349 and 269, respectively.

number of MNR-1:: GFP and DMA-1::RFP were pull-downed in the control group in the absence of SAX-7::HA (*Figure 10A*). Importantly, in the presence of LECT-2::FLAG, SAX-7::HA required 40-folds less input than the sample without LECT-2 to pull down a similar amount of MNR-1 and DMA-1 (*Figure 10A–B*). The expression amount of DMA-1::RFP or MNR-1::GFP were comparable between two samples (*Figure 9—figure supplement 1*). Our result suggests that LECT-2 increases the binding efficiency between SAX-7, MNR-1 and DMA-1 by roughly 40-folds in this experimental setup. Together, our evidence strongly suggests that LECT-2 promotes the formation of the receptor-ligand complex by increasing the binding efficiency between SAX-7, MNR-1 and DMA-1.

## Discussion

Precise dendritic morphogenesis and guidance rely on the interaction between receptors on the dendritic surface and ligands from their growth environment. In this study, we showed that LECT-2 is a novel and essential component of a multi-protein receptor-ligand complex that underlies targeting and patterning of PVD dendrites.

### LECT-2 acts both locally and globally to guide and pattern PVD dendrites

Previous studies revealed that SAX-7 forms specific patterns on the epidermis where PVD dendrites arborize (*Dong et al., 2013*; *Liang et al., 2015*; *Salzberg et al., 2013*). However, the PVD dendrites do not simply follow wherever SAX-7 levels are high. For example, SAX-7 is enriched on the epidermal-seam cell junctions near the primary dendrite of the PVD neuron (*Chen et al., 2001*; *Dong et al., 2013*), yet PVD dendrites largely ignore the SAX-7 there and extend towards the muscle cells. Thus, additional factors likely guide the PVD innervation of dendrites toward the muscles. Our results support that LECT-2 acts locally to promote the formation of the terminal branches of PVD on muscles. First, loss of *lect-2* completely abolishes the muscle targeting of PVD dendrites. Second, *lect-2* is expressed in muscles and functions as a local-acting guidance cue to guide the growth of terminal branches. Third, seam cell expressed LECT-2 fails to restore muscle targeting, but instead targets PVD dendrites onto seam cells in *lect-2* mutants.

While LECT-2 acts locally to promote 4° dendrites formation, our data also showed that it could diffuse in the body cavity and act as a permissive cue for the patterning of 2° and 3° dendrites. Interestingly, overexpression of *lect-2* from several different tissues resulted in the near complete rescue of the mutant phenotype in dendrite shape. These results argue that when overexpressed, the cellular source of *lect-2* is not critical for shaping the dendritic arbors, suggestive of a permissive function of *lect-2*. While the interpretation of these experiments should be cautioned due to overexpression, they do suggest that additional unidentified mechanisms exist to specify the muscles rather than the seam cells as the innervating targets of PVD dendrites.

### LECT-2 acts together with co-ligands SAX-7 and MNR-1 to activate the neuronal receptor DMA-1

Both genetic and biochemical evidence demonstrates that LECT-2 functions together with SAX-7 and MNR-1 on the epidermis to activate the dendritic receptor DMA-1. First, loss of *lect-2* causes dendritic targeting and patterning defects indistinguishable from those of *sax-7* and *mnr-1* mutants. Second, the PVD dendritic targeting and patterning defects in the double mutants between *lect-2* and *sax-7/mnr-1/dma-1* are not further enhanced, suggesting that all four genes function in the same genetic pathway. Third, LECT-2 interacts with SAX-7, and forms a multi-protein complex that likely includes SAX-7, MNR-1 and DMA-1. Fourth, LECT-2 greatly enhances the binding efficiency between SAX-7, MNR-1 and DMA-1. These results are consistent with a model in which every component of this signaling receptor complex is essential for its function (*Figure 10C*).

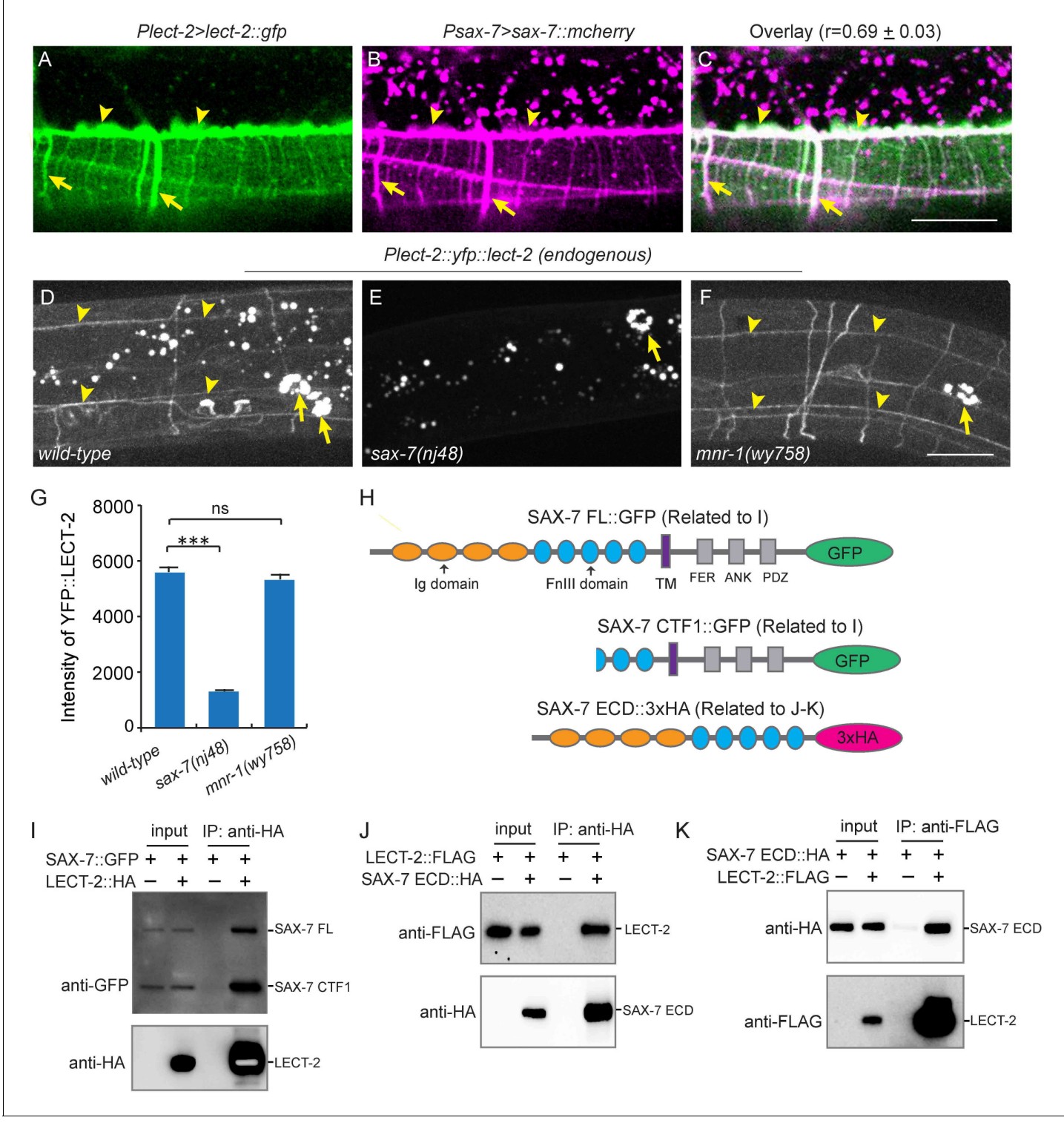

**Figure 8.** LECT-2 interacts with SAX-7. (**A–C**) Confocal images showing co-localization between LECT-2::GFP (green) and SAX-7::mCHERRY (magenta). Arrow heads: tertiary line along the border of outer body wall muscles; Arrows: quaternary stripes where quaternary dendrites grow along. Scale bar: 20 μm. The Pearson's coefficient index was measured from 12 animals. Data are represented as mean ± SEM. (**D–F**) Confocal images showing the patterns of endogenously expressed YFP::LECT-2 in (**D**) *wild-type*, (**E**) *sax-7(nj48)* and (**F**) *mnr-1(wy758)* mutant animals. Scale bar: 20 μm. (**G**) Quantifications of intensity of YFP::LECT-2 along the border of outer body wall muscles. Intensity of background was measured in the region outside of worms and subtracted. One way ANOVA with the Dunnett's test was used for statistical analysis. ***p<0.0001. ns: not significant. Data are represented as mean ± SEM. 10 animals were quantified for each genotype. (**H**) Schematics of tagged full length or truncated SAX-7 expressed in *Drosophila* S2 cells for co-

*Figure 8 continued on next page*

*Figure 8 continued*

immunoprecipitation (co-IP) experiments. FL: full length. Ig: immunoglobulin domain. FnIII: fibronectin domain III. TM: transmembrane domain. FER: conserved FERM domain. ANK: Ankyrin-binding domain. PDZ: PDZ domain. CTF1: C-terminal fragment 1 (generated by cleavage through a putative furin cleavage site in the third FnIII domain). ECD: extracellular domain. (I) Western blot images showing co-IP between LECT-2::HA and SAX-7::GFP. IP: immunoprecipitation. WB: western blot. (J–K) Western blot images showing co-IP between LECT-2::FLAG and SAX-7 ECD::HA. Note that in these experiments a mutant form of SAX-7 ECD::HA construct with furin cleavage site mutations was utilized.

The following figure supplements are available for figure 8:

**Figure supplement 1.** Transcription of *lect-2* is largely not affected by loss of *sax-7*.

**Figure supplement 2.** The last two FnIII domains of SAX-7 are sufficient to interact with LECT-2.

## LECT-2, SAX-7 and MNR-1 generate a novel combinatorial dendritic guidance code

In contrast to known axon guidance receptors that are often activated by a single ligand, the activation of the DMA-1 receptor appears to require the presence of all three ligands—SAX-7, MNR-1 and LECT-2. It is interesting to note that the three extrinsic ligands are made by two different tissues: muscles and epidermis, and that PVD dendrite arborizes precisely between these tissues. The exclusive expression of MNR-1 by the epidermal cells ensures that PVD dendrites grow along the epidermis (*Dong et al., 2013*; *Salzberg et al., 2013*). On the other hand, SAX-7 forms specific patterns on the epidermis to specify the location of 3° dendrites and determine the regular spacing of 4° dendrites (*Dong et al., 2013*; *Liang et al., 2015*; *Salzberg et al., 2013*). In this study, we showed that muscle secreted LECT-2 acts locally to ensure muscle innervation by the terminal branches of PVD neurons. DMA-1 acts as a coincidence detector, which is only activated when LECT-2, SAX-7 and MNR-1 are all present. This combinatorial code results in the precise targeting and patterning of PVD dendrites (*Figure 10C*).

During development, guidance decisions are often made by a combination of molecular cues. For example, *Drosophila* embryonic CNS neurons use two semaphorins to establish specific connectivity. Sema-2a acts as a repulsive cue, while Sema-2b functions as an attractive cue. Both secreted semaphorins act through the same neuronal receptor PlexB, suggesting that different axon guidance cues can converge on the same receptor (*Wu et al., 2011*). In contrast, the same guidance cue can also elicit different responses by activating different receptors. The classic example is netrin/UNC-6, which can attract growth cones through UNC-40/DCC and repel neurons expressing UNC-5 (*Chan et al., 1996*; *Colamarino and Tessier-Lavigne, 1995*; *Hedgecock et al., 1990*; *Kennedy et al., 1994*; *Leonardo et al., 1997*; *Serafini et al., 1994*; *Wadsworth et al., 1996*). In the above-mentioned cases, a single ligand is sufficient to bind to and activate its receptor. The integration of guidance information from multiple cues or the differential response to a single cue likely occurs downstream of guidance receptors. In contrast to these published examples, LECT-2, SAX-7 and MNR-1 converge at the level of the DMA-1 receptor and form an 'and' gate for the activation of the receptor. Thus, the combinatorial dendritic guidance code reported here represents a new type of mechanism to integrate diverse cues for morphogenesis decisions.

The PVD neuron responds to multiple sensory modalities including harsh touch, cold, elevated temperature, hyperosmolarity and possibly muscle contraction (*Albeg et al., 2011*; *Chatzigeorgiou et al., 2010*; *Mohammadi et al., 2013*; *Smith et al., 2013*). It is conceivable that the PVD dendrite's attachment to the epidermis is required for touch related functions. It is likely that the precise location of PVD dendrites between epidermis and muscle cells is essential for its function as a proprioceptive sensory neuron. Therefore, the elaborate developmental program achieves specific dendrite morphogenesis at desired locations to facilitate PVD's sensory functions. In mammals, there are many examples of precise patterning of dendritic morphogenesis in the central and peripheral nervous system, including the precise innervation of specialized peripheral sensory organs by sensory neurons (*Cheret et al., 2013*). Future studies are needed to determine whether a similar coding mechanism is used to achieve precise dendrite innervation in mammals. LECT2, the vertebrate homolog of LECT-2, was discovered based on its activity as a neutrophil chemotactic factor (*Yamagoe et al., 1996*). Known neuronal guidance molecules such as Slit have been

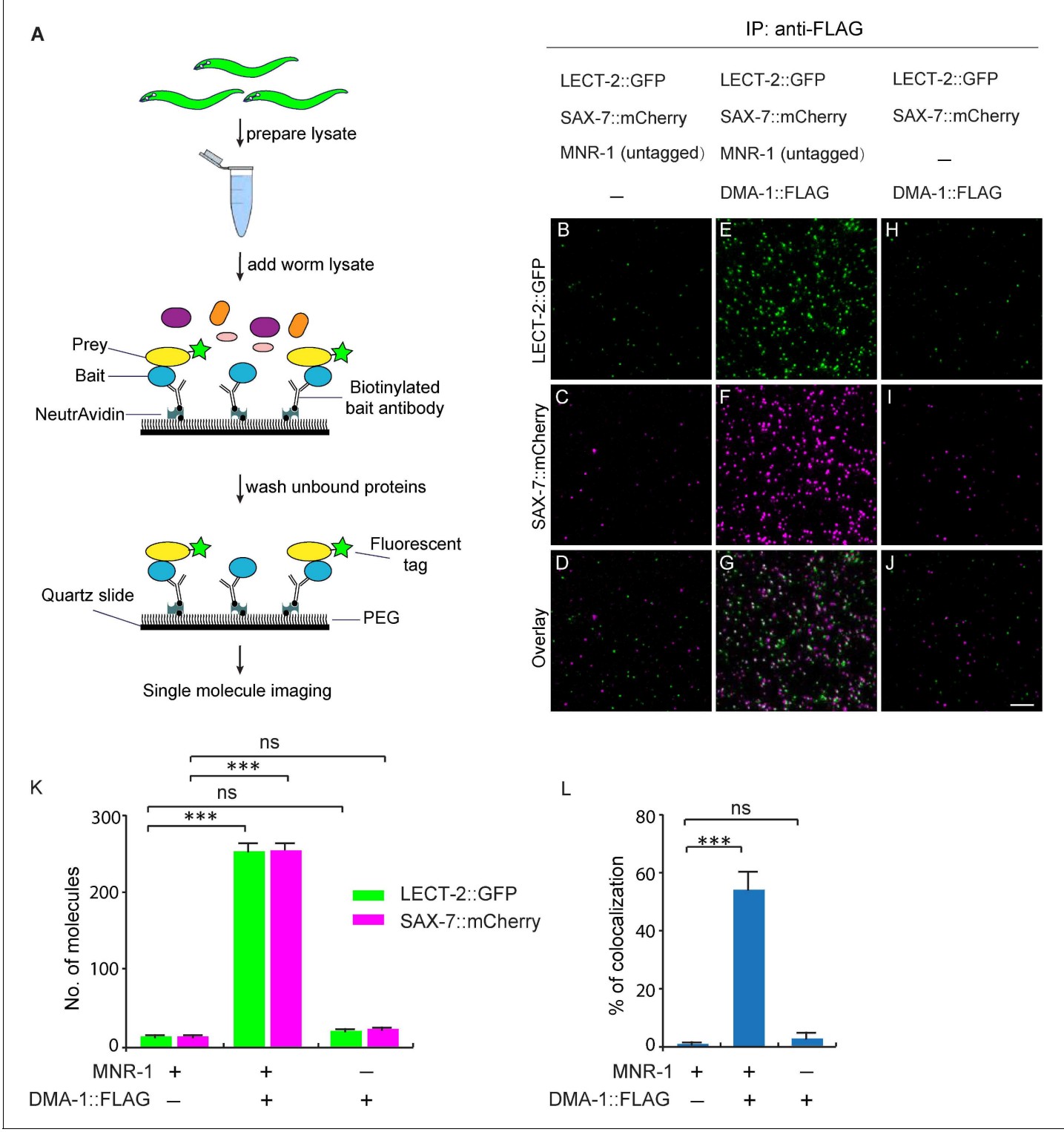

**Figure 9.** LECT-2 forms a multi-protein complex with SAX-7, MNR-1 and DMA-1. (**A**) A cartoon showing how the single-molecule pull-down (SiMPull) assays were performed. (**B–J**) Representive fluorescent images showing the results of SiMPull assays. Specific pull-down of LECT-2 (green) and SAX-7 (magenta) were only observed from lysate co-expressing DMA-1::FLAG and MNR-1 (**E–G**), but not lysates without DMA-1::FLAG(B-D) nor MNR-1(H–J). Scale bar: 5 µm. (**K**) Quantification of 10 different regions of the imaging surface for each group. (**L**) Single-molecule colocalization between LECT-2:: GFP and SAX-7::mCherry. The number of molecules where colocalization occurred divided by the total number of mCherry molecules was calculated as percentage of colocalization. 10 different regions of the imaging surface were imaged and quantified. One way ANOVA with the Dunnett's correction was used for statistical analysis. ***p<0.0001. Data are represented as mean ± SEM.

*Figure 9 continued on next page*

eLIFE Research article

Developmental Biology and Stem Cells | Neuroscience

*Figure 9 continued*

The following figure supplement is available for figure 9:

**Figure supplement 1.** Prey proteins are expressed at similar levels in different groups.

shown to regulate leukocyte chemotaxis through the CXCR4 chemokine receptor (*Wu et al., 2001*). Hence, the acquired immune system utilized ancient molecules from the developmental nervous system to coordinate the polarization and movement of cells.

## Materials and methods

### Strains and genetics

N2 Bristol was used as the wild-type strain. Worms were grown on OP50 *E. coli* seeded nematode growth medium plates at 20°C or room temperature, following standard protocols (*Brenner, 1974*). The mutant alleles used in this study were *lect-2(wy935)*, *lect-2(wy953)*, *lect-2(ok2617)*, *sax-7(nj48)*, *mnr-1(wy758)* and *dma-1(wy686)*. For details and complete lists of strains see *Supplementary file 1*.

### Constructs and transgenes

Most of the plasmid constructs were generated in pSM delta vector (a derivative of pPD49.26 with additional cloning sites). For details and complete lists of plasmids see *Supplementary file 1*. *lect-2* cDNA was amplified from a mix-stage worm cDNA library (kindly provided by Dr. Kota Mizumoto) using primers oWZ477 (gc ggcgcgcc atgcatctgagaaccttgcattttc) and oWZ452 (gc ggtacc ttagaatactggaaagttcggag). Worm-codon optimized Cre was amplified from pSR40 (kindly provided by Dr. Sander van den Heuvel) (*Ruijtenberg and van den Heuvel, 2015*). *Plect-2(3.4 kb)::lect-2* genomic DNA was amplified from fosmid WRM0634aG06 using primers oWZ447 (gc gcatgc ttacaagcattgacactccctt) and oWZ450 (cg cccggg ttagaatactggaaagttcggag). *Plect-2(1.5 kb)::lect-2* genomic DNA was amplified similarly, except that oWZ448 (gc ggcgcgcc cagtatgaaaaaaaaaggaaatttctcagaatcc) was used as the forward primer. pWZ347 was a derivative of pCFJ909 (miniMos vector, kindly provided by Dr. Christian Frøkjær-Jensen) with additional cloning sites and two loxp sites flanked the multiple cloning sites. *Plect-2(1.5 kb)::lect-2::SL2::mcherry::unc-54* was cut from a pSM delta-based intermediate construct and inserted into pWZ347 and used to make the single copy transgene *wyTi3*.

A CRISPR/Cas9-mediated homologous recombination method was used to insert yfp right after the sequence encoding the predicted signal peptide (1–20 amino acids) of *lect-2* (*Armenti et al., 2014*; *Dickinson et al., 2013*). pWZ374 (*pU6>lect-2-sgRNA#1*, target sequence: taatttacaggtcagacat) and pWZ375 (*pU6>lect-2-sgRNA#2*, target sequence: aatttacaggtcagacatt) were made by replacing the target sequence in pBHC1084 (kindly provided by Dr. Baohui Chen) using a phosphorylated primer-based quick-change method. pWZ374 and pWZ375 were co-injected together with pJW1259 (*Peft-3>cas9*, kindly provided by Dr. Jordan Ward), a PCR product (left arm-yfp-*reverse Cbr-unc-119*-right arm, repair template), P*myo-2>mcherry*, P*myo-3>mcherry* and P*odr-1>rfp* into *unc-119(ed4)* worms. 45 bp left homology arm and 42 bp right homology arm were designed in the primers to amplify yfp from pJN601 (kindly provided by Dr. Jeremy Nance) (*Paix et al., 2014*). A loxp-flanked reverse Cbr-*unc-119* rescue fragment was embedded in an intron of yfp.

Transgenes expressed from extrachromosomal arrays were generated using standard gonad transformation by injection (*Mello and Fire, 1995*). P*myo-2>mcherry*, P*myo-3>mcherry*, P*odr-1>gfp*, P*odr-1>rfp*, P*unc-122>rfp* or *unc-119(+)* plasmid was injected at 2–40 ng/μl as co-injection markers.

To make mosaic transgenic worms in which only a few muscles expressed LECT-2, plasmid pWZ299 (P*hlh-1>lect-2::SL2::mcherry*) was injected into *lect-2(ok2617)*; *wyIs592 (ser2prom3>myr-gfp)* worms at 0.2 ng/μl to form extrachromosomal arrays. pWZ350 (P*nhr-81>mcherry*, 20 ng/μl), P*odr-1>gfp* (30 ng/μl) and P*myo-2>mcherry* (2 ng/μl) were co-injected as co-injection markers. In two independent lines, mosaic expression of mCherry was observed in muscles: most but not all the muscle cells expressed the transgenes. A small fraction of worms only expressed LECT-2::SL2::

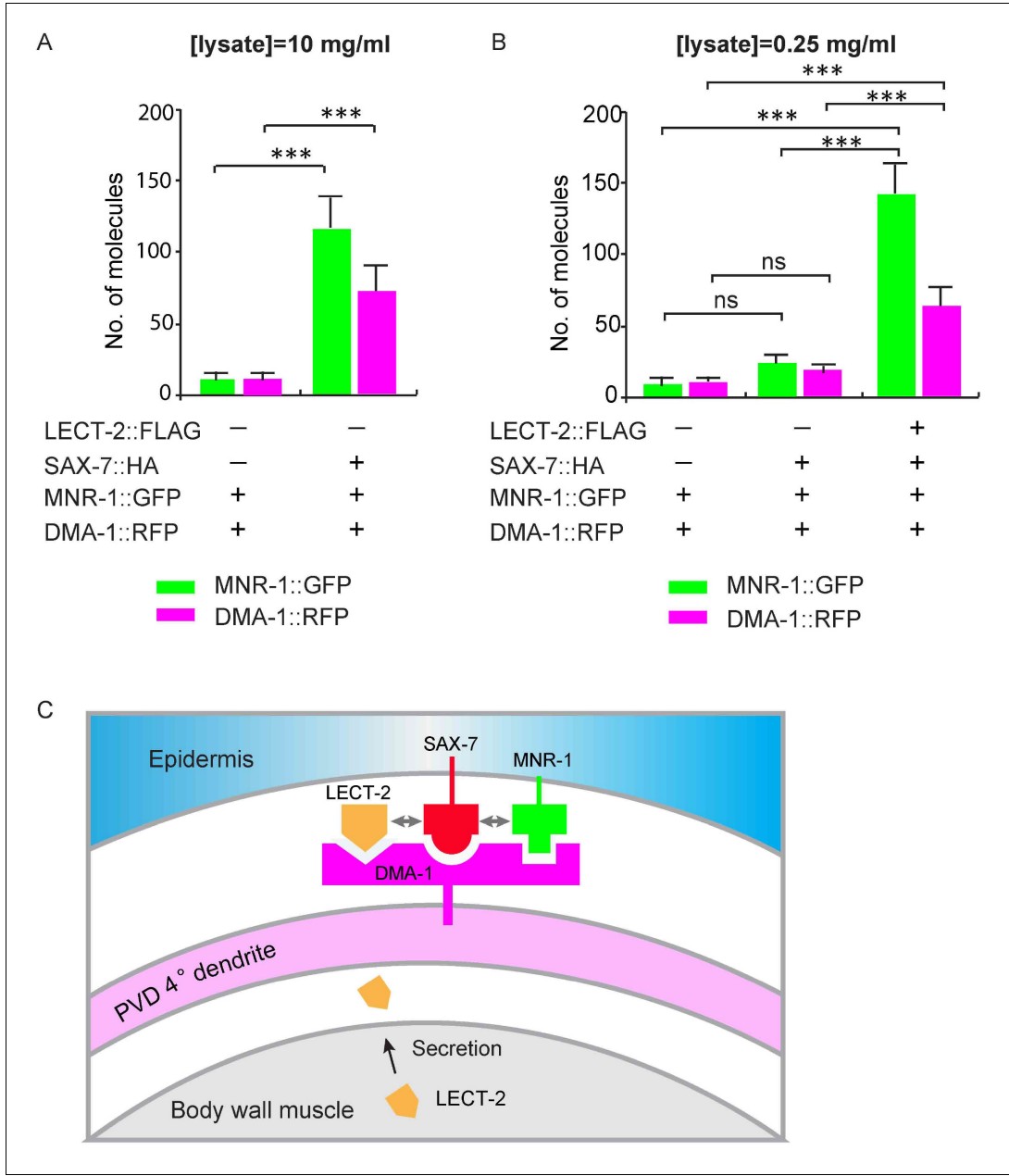

**Figure 10.** LECT-2 enhances the binding among SAX-7, MNR-1 and DMA-1 in vitro. (A–B) *Drosophila* S2 cells expressing DMA-1::RFP and MNR-1::GFP, with or without co-expression of SAX-7::HA and LECT-2::FLAG, were used to detect protein interactions by SiMPull. Lysates were applied to a chamber coated with anti-HA antibody. Different lysate concentrations (10 and 0.25 mg/ml for A and B, respectively) were used in order to achieve an optimum density of fluorescent proteins. Specific pull-down of DMA-1 and MNR-1 were observed from lysate co-expressing SAX-7 HA, but not lysates without SAX-7::HA. The addition of LECT-2::FLAG required 40-folds less concentration of input than the sample without LECT-2 co-expression, to achieve similar MNR-1 and DMA-1 pull-down outputs. 10 and 13 different regions of the imaging surface were imaged and quantified for the two groups shown in (A). 5, 5 and 12 different regions of the imaging surface were imaged and quantified for the three groups shown in (B). One way ANOVA with the Tukey correction was used for statistical analysis. ***$p<0.0001$. ns: not significant. Data are represented as mean ± SEM. (C) A cartoon showing that LECT-2 is secreted from body wall muscles and captured by SAX-7 on the epidermis. SAX-7 also directly binds to MNR-1. LECT-2, SAX-7 and MNR-1 form a combinational dendritic guidance cue to bind and activate dendritic receptor DMA-1 to regulate targeting and morphogenesis of PVD dendrites.

*Figure 10 continued on next page*

*Figure 10 continued*

The following figure supplement is available for figure 10:

**Figure supplement 1.** Adding LECT-2::FLAG increases the aggregation formation between DMA-1::RFP and SAX-7::GFP-MNR-1::GFP -expressing S2 cells.

mCherry in a few muscles based on the mCherry signal under the fluorescent microscope. These animals were selected for further analyses.

## Isolation and mapping of *lect-2 (wy935)* and *lect-2 (wy953)* mutants

The *wy935* and *wy953* alleles were isolated from an F2 semi-clonal screen of 3000 haploid genomes in the *dma-1(wy908); wyIs592* [*ser2prom3>myr-gfp*] genetic background (*Dong et al., 2016*). Worms were mutagenized with 50 mM ethyl methane sulfonate (EMS). SNP mapping and transgene rescue experiments were performed using standard protocols as described below (*Davis et al., 2005*; *Mello and Fire, 1995*). Specifically, *wy935* allele was used to map the gene affected by this mutation using standard SNP mapping method and was mapped between −1.9 and −0.27 of LGII. A deficiency strain *ccDf5* (−5.15 to −0.92? of LGII was deleted) was used to do deficiency mapping and further narrowed down the mutation to a region between −0.92 and −0.27. Twenty-four fosmids which covered the above region were selected and injected into *wy935; wyIs592* worms. Only fosmid WRM0634aG06 was found to be able to fully rescue the dendrite morphogenesis defect of *wy935* mutants. Five partially overlapped PCR fragments were amplified from WRM0634aG06 and injected individually to test their rescue ability. PCR fragment #2, which only contained ORF K05F1.5, could fully rescue *wy935* mutants. Genomic DNA of K05F1.5 was amplified from *wy935* and *wy953* worms for sequencing. The *wy935* allele carries a mis-sense G-to-A point mutation flank by sequences TTCATTGTTG and AATTGATGAT. The *wy953* allele carries a mis-sense G-to-A point mutation flanked by sequences AGAATTGAGG and AACCGGGCAG.

## Imaging and quantification of dendritic branching

Mid-L4 to young adult stage hermaphrodite animals were anesthetized using 10 mM levamisole in M9 buffer, mounted on 2% agar pads and imaged using a Zeiss LSM710 confocal microscope (Carl Zeiss) with a Plan-Apochromat 40x/1.3 NA objective (for most images showed in this study) or a spinning disk confocal microscope with a 40x/1.3 NA or 63x/1.4 NA objective (for images showed in *Figure 4C*, *Figure 8A–F*, *Figure 1—figure supplement 1* and *Figure 8—figure supplement 1*). Z stacks and maximum-intensity projections were generated using ImageJ (NIH) or ZEN 2009 software. The imaging was not done by an experimenter blind to the experimental condition. Colocalization analysis (*Figure 8C*) and fluorescence intensity (*Figure 8G*) were quantified using plugins of ImageJ (NIH).

For quantifications showed in *Figures 1*, *2*, *3*, *4* and *5* and *Figure 6—figure supplement 1*, all branches within 100 μm of the primary dendrite anterior to the PVD cell body were counted. Every lateral branch from the primary dendrite was scored as a 2° dendrite. 'T' shaped branches along the border of outer body wall muscles were scored as 3° dendrites, and all branches derived from 3° dendrites were scored as 4° dendrites. Statistical comparisons were conducted using Student's t-test (to test for differences between two groups) or one-sided ANOVA with the Dunnett's test or Tukey correction (to test for differences between three or more groups).

## S2 aggregation and Co-IP assays

S2 aggregation assays were performed as previously described with some modifications (*Dong et al., 2013*). Briefly, *Drosophila* S2 cells were cultured in Schneider's insect medium (Sigma) according to the manufacturer's description and transfected using Effectene (Qiagen). Three days after transfection, S2 cells were washed with PBS three times. Mixed cells were re-suspended in either S2 medium collected from non-transfected cells or *Pactin>lect-2::3xflag* transfected cells and rotated at 30 rpm at room temperature. 3 μl of each mixture was spotted on glass slides for imaging and quantification after 0 min, 10 min and 40 min. The experiments have been repeated for at least three times and consistent results were obtained.

For co-IP experiments for two secreted proteins, mediums were collected 3 days after transfection. Anti-HA or Anti-FLAG beads (Sigma) were used to incubate with the mediums for 2 hr at 4°C with rotation. The beads were washed with cell lysis buffer three times. Proteins were eluted at 65°C using 2% SDS elution buffer and detected using Western blot analysis with mouse antibody to HA (1:1000, Roche), mouse antibody to FLAG (1:2000, Sigma), and HRP-conjugated goat antibodies to mouse (1:20,000, Jackson Immuno Research). For co-IP experiments between transmembrane proteins and secreted proteins, S2 cells were collected 3 days after transfection and lysed in the cell lysate buffer (for transmembrane proteins). Media were separately collected from non-transfected cells and *Pactin>lect-2::3xflag* (or *Pactin>lect-2::3xHA*) transfected cells and mixed with cell lysates for transmembrane proteins. Other steps were similar to those described above. Other antibodies used in this study include rabbit antibody to Myc (1:2000, Santa Cruz Biotechnology), mouse antibody to GFP (1:2000, Roche), and HRP-conjugated goat antibodies to rabbit (1:20,000, Jackson Immuno Research). The experiments have been repeated for at least three times and consistent results were obtained.

### Single-molecule pull-down assay (SiMPull)

*Drosophila* S2 cells over-expressing DMA-1::RFP, MNR-1::GFP, SAX-7::HA and LECT-2::FLAG were pelleted and lysed in lysis buffer (50 mM HEPES pH 7.7, 150 mM NaCl, 2 mM MgCl2, 1 mM EDTA pH 8.0, 1% Triton X-100 with protease inhibitors) at 4°C for 1 hr. After centrifugation 16000 g 15 min, supernatants were collected and measured by BCA assay for total protein concentration (Thermo Fisher Scientific).

*C. elegans* grown on twenty 6 cm dishes were collected and washed, then dropped in liquid nitrogen to form "worm pearls'. Worm pearls (300 mg wet weight) were thaw in 150 µl lysis buffer (50 mM HEPES pH 7.7, 50 mM KCl, 2 mM MgCl2, 250 mM Sucrose, 1 mM EDTA pH 8.0, with protease inhibitors). After briefly sonicate on ice (5' pulse with 59" pause, 5 cycles) to break cuticle, 100 mM NaCl and 1% Triton X-100 were added into solution and samples were rotated at 4°C for 1 hr. After centrifugation, supernatants were collected and measured by BCA assay for total protein concentration.

Worm and S2 cell lysates were adjusted by lysis buffer to desired concentrations in order to achieve an optimum density of fluorescent proteins on the surface of SiMPull slides (100–400 molecules in a 2000 $\mu m^2$ imaging area). Briefly, worm lysates with 7 µg/ml total protein concentration were applied onto quartz slides coated with biotinylated anti-GFP or anti-RFP antibodies (Rockland immunochemicals) for LECT-2::GFP or SAX-7::mCherry pull down, and worm lysates with 10 mg/ml total protein concentration were injected into slides coated with biotinylated anti-FLAG antibody (Sigma Aldrich) for DMA-1::FLAG pull down. S2 cell lysates with 10 mg/ml or 250 µg/ml total protein concentration were used on slides coated with biotinylated anti-HA antibody (Abcam) for SAX-7::HA pull down, and S2 cell lysates with 100 µg/ml total protein concentration were used for MNR-1::GFP and DMA-1::RFP pull down.

Proteins immobilized on the slides were visualized by a TIRF microscope equipped with excitation laser 488 nm (GFP) and 561 nm (mCherry or RFP), and DV2 dichroic 565dcxr dual-view emission filters (520/30 nm and 630/50 nm). In all cases, mCherry fluorescence was collected first, followed by GFP fluorescence at the same position. 5–13 different regions of the imaging surface were imaged and quantified. Single-molecule colocalization between GFP and mCherry was performed using a method described previously (*Jain et al., 2011*). The mCherry and GFP molecules within a 2-pixel distance (~300 nm) were considered as co-localized. The number of molecules where colocalization occurred divided by the total number of mCherry molecules was calculated as overlap percentage.

### Acknowledgements

This work was supported by the Howard Hughes Medical Institute and the National Institute of Neurological Disorders and Stroke (1R01NS082208 to KS) and the National Heart, Lung, and Blood Institute (R01HL127764-01 to YKX). AS is an American Heart Association postdoctoral fellowship recipient. Some strains were provided by the *C. elegans* Gene Knockout Consortium and the CGC (funded by the NIH Office of Research Infrastructure Programs (P40 OD010440)). We thank Drs. Erik Jorgensen, Christian Frøkjær-Jensen, Jeremy Nance, Jordan Ward, Liqun Luo, Kota Mizumoto, Baohui Chen and Sander van den Heuvel for kindly sharing reagents and Dr. Engin Özkan for discussion.

We also thank Cen Gao for technical assistance, and Drs. Peri Kurshan and Claire Richardson for thoughtful comments on the manuscript. After we submitted this manuscript to Elife, an independent study by Dr. Hannes Bulow's group on lect-2 and dendrite morphogenesis was submitted and published in Current Biology.

## Additional information

### Competing interests

KS: Reviewing editor, *eLife*. The other authors declare that no competing interests exist.

### Funding

| Funder | Grant reference number | Author |
|---|---|---|
| Howard Hughes Medical Institute | | Wei Zou<br>Xintong Dong<br>Madina Tugizova<br>Kang Shen |
| National Institute of Neurological Disorders and Stroke | 1R01NS082208 | Wei Zou<br>Xintong Dong<br>Madina Tugizova<br>Kang Shen |
| National Heart, Lung, and Blood Institute | R01HL127764-01 | Ao Shen<br>Yang K Xiang |
| American Heart Association | | Ao Shen |

The funders had no role in study design, data collection and interpretation, or the decision to submit the work for publication.

### Author contributions

WZ, Conception and design, Acquisition of data, Analysis and interpretation of data, Drafting or revising the article, Contributed unpublished essential data or reagents; AS, YKX, Acquisition of data, Analysis and interpretation of data, Drafting or revising the article, Contributed unpublished essential data or reagents; XD, Performed the S2 aggregation experiments, Contributed unpublished essential data or reagents; MT, Isolated wy953 allele and helped WZ with part of mapping and cloning experiments, Contributed unpublished essential data or reagents; KS, Conception and design, Analysis and interpretation of data, Drafting or revising the article

### Author ORCIDs

Kang Shen, http://orcid.org/0000-0003-4059-8249

## Additional files

### Supplementary files

• Supplementary file 1. Strains and plasmids used in this study. (a) Strains used in this study. (b) Plasmids used in this study.

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
