## [Decision Letter]

Thank you for submitting your article "A multi-protein receptor-ligand complex underlies combinatorial dendrite guidance choices in *C. elegans*" for consideration by *eLife*. Your article has been favorably evaluated by Eve Marder (Senior Editor) and three reviewers, one of whom, Graeme W Davis (Reviewer #1), is a member of our Board of Reviewing Editors. The following individuals involved in review of your submission have agreed to reveal their identity: David M Miller III (Reviewer #2); Joshua M Kaplan (Reviewer #3).

The reviewers have discussed the reviews with one another and the Reviewing Editor has drafted this decision to help you prepare a revised submission.

Summary:

This paper reports a novel role for a diffusible cue in dendritic morphogenesis. The authors performed a forward genetic screen in *C. elegans* for mutations that disrupt the precise, repetitious pattern of dendrite arborization in the body wall of this organism. The authors found two mutations in a single gene, both genetically behaving like the gene deletion, indicative of these being loss of function or null mutations. Dendrites are severely disrupted. The authors perform a formal genetic analysis, proving that the DMA2 gene is causal for the observed phenotype. DMA2 encoded the worm homologue of LECT2. Further genetic analyses implicate this gene is a previously established signaling system that includes the worm homologue of L1CAM. A strength of the paper is the magnitude and penetrance of the phenotype and the genetic analyses inclusive of rescue. More specifically, the authors show that DMA-2 is secreted from body muscle cells to promote the outgrowth of terminal ("quaternary") branches in a region sandwiched between muscle and overlying epidermis. A combination of biochemical and cell aggregation experiments are consistent with a model in which DMA-2 enhances the interaction of 3 additional components, SAX-7/L1, MNR-1 and DMA-1 which the authors have previously shown are required for promoting adherence of PVD dendrites to the epidermis. The results reported firmly establish DMA-2 as a new guidance molecule for PVD dendrite development. However, there are remaining concerns that should be addressed, including a more quantitative and thorough analysis of the dendritic arborization phenotype to more precisely define the nature and effects of the signal that has been identified and a more careful interpretation of the data that are being currently presented.

Major Revision:

1) The results shown in Figure 3, Figure 4 and Figure 5 suggest that DMA-2 might also function as a permissive signal to promote outgrowth of secondary and tertiary branches. In Figure 3, targeted ablation of DMA-2 expression in muscle (with a muscle-specific cre driver) significantly reduces the number of quaternary branches but secondary and tertiary branches appear largely normal. This result stands in contrast to the *dma-2* mutant phenotype which shows highly aberrant secondary and tertiary branch defects in addition to the absence of quaternary branches (Figure 1). Although PVD quaternary branch outgrowth is selectively rescued in a *dma-2* mutant for a single DMA-2-expressing body muscle cell (Figure 4), secondary and tertiary branch outgrowth in other regions is also apparently restored. Finally, Figure 5 shows aberrant innervation of lateral seam cells with local over expression of DMA-2 but again, secondary and tertiary branch morphology is also partially rescued in this case despite the lack of DMA-2 expression in muscle. One possibility is that ectopic expression of DMA-2 in other cell types (i.e., non-muscle) may rescue secondary and tertiary branch outgrowth but not the quaternary branch defect in *dma-2* mutants. Indeed, it would be straightforward to test the idea that DMA-2 could also function as a permissive cue by expressing it in another cell type (e.g., ventral cord motor neurons). Importantly, but not the only experiment that should be performed, does seam cell expressed DMA-2 rescue 2^o^ and 3^o^ branch defects? These experiments should be conducted.

2) Prior papers showed that *dma-1, mnr-1*, and *sax-7* mutants have defects in 2^o^, 3^o^, and 4^o^ dendrite branches. Do *dma-2* mutants also have 2^o^ and 3^o^ branching defects? The authors should quantify these effects rigorously and the authors should modify the text and discussion to describe these results and interpretations more clearly. This should be inclusive of the mutant phenotype, rescue and overexpression experiments.

3) The distribution of the DMA-2/GFP (Figure 3—figure supplement 2C-H) and its rescuing activity (Figure 3—figure supplement 2B) should be quantified. Is DMA-2/GFP's localization significantly altered in *sax-7* or *mnr-1* mutants?

4) Many results were not quantified, and sample sizes were only summarized (e.g. "at least 12 animals") for those that were quantified. Please quantify all reported results and please indicate specific sample sizes for every genotype in each figure.

5) The coIP data are nice but do not document a single complex of all 4 proteins. The coIPs also do not convincingly show that DMA-2 alters the affinity of the complex for the other proteins. More specifically, there is a failure to replicate a previous result showing that the receptor complex can form in the absence of DMA-2. The results shown here suggest that DMA-2 is required (and in that sense clearly "enhances" interaction). It should be possible to show a quantitative dependence on DMA-2. Beyond this, the interpretation of the biochemistry should be more circumspect.

6) The Discussion says: "It is interesting to note that the three ligands from two tissues contain distinct information for PVD dendrite patterning." If the mutants have identical phenotypes, this can't be true or, at a minimum, is very confusing to the reader. The authors should address this issue experimentally or through modification of the text.

*Reviewer #1:*

The authors have performed a forward genetic screen in *C. elegans* for mutations that disrupt the precise, repetitious pattern of dendrite arborization in the body wall of this organism. The authors found two mutations in a single gene, both genetically behaving like the gene deletion, indicative of these being loss of function or null mutations. Dendrites are severely disrupted. The authors perform a formal genetic analysis, proving that the DMA2 gene is causal for the observed phenotype. DMA2 encoded the worm homologue of LECT2. Further genetic analyses implicate this gene is a previously established signaling system that includes the worm homologue of L1CAM. The phenotypes being described are extremely strong, robust and all rescue experiments are equally clean and clear. The remaining genetic analyses are performed to a very high standard, characteristic of the Shen laboratory. The weakest link in the paper is the final figure (Figure 7) regarding the biochemical analyses. However, in my view, these data are reasonably interpreted and help provide a conceptual model upon which future experiments can be performed. My only substantive criticism is that the paper is written for *C. elegans* neurobiologists and less for a general reader. It would be nice to have gene names linked to the mammalian orthologues throughout the paper, and for the text to touch base, routinely with the larger literature, not just the emerging signaling system in this organism. This should be an easy fix that would make the paper more generally accessible and would make the findings more clearly relevant to the biology of other systems, which it certainly is.

*Reviewer #2:*

This paper reports a novel role for a diffusible cue in dendritic morphogenesis. The authors exploited the stereotypically branched architecture of the *C. elegans* PVD sensory neuron to detect mutations in a conserved protein, (named DMA-2 here), in a forward genetic screen. The authors show that DMA-2 is secreted from body muscle cells to promote the outgrowth of terminal ("quaternary") branches in a region sandwiched between muscle and overlying epidermis. A combination of biochemical and cell aggregation experiments are consistent with a model in which DMA-2 enhances the interaction of 3 additional components, SAX-7/L1, MNR-1 and DMA-1 which the authors have previously shown are required for promoting adherence of PVD dendrites to the epidermis. The authors conclude that DMA-2 functions strictly as an instructional cue to promote quaternary branch outgrowth. Although experiments reported here strongly support this hypothesis, additional results shown in Figure 3, Figure 4 and Figure 5 suggest that DMA-2 might also function as a permissive signal to promote outgrowth of secondary and tertiary branches. In Figure 3, targeted ablation of DMA-2 expression in muscle (with a muscle-specific cre driver) significantly reduces the number of quaternary branches but secondary and tertiary branches appear largely normal. This result stands in contrast to the *dma-2* mutant phenotype which shows highly aberrant secondary and tertiary branch defects in addition to the absence of quaternary branches (Figure 1). Perdurance seems to be an unlikely explanation since the cre driver (*hlh-1*) is expressed in embryonic muscle whereas the PVD phenotype is scored much later in development at the L4 larval stage. A similar result is also evident in a mosaic expression experiment reported in Figure 4. Although PVD quaternary branch outgrowth is selectively rescued in a *dma-2* mutant for a single DMA-2-expressing body muscle cell (Figure 4), secondary and tertiary branch outgrowth in other regions is also apparently restored. Finally, Figure 5 shows aberrant innervation of lateral seam cells with local over expression of DMA-2 but again, secondary and tertiary branch morphology is also partially rescued in this case despite the lack of DMA-2 expression in muscle. The authors need to address potential permissive role of DMA-2 with additional experiments (see below). Otherwise, experiments reported here are convincing, the writing is clear, images are compelling and the findings offer a new understanding of the molecular mechanisms that drive dendrite morphogenesis.

1) As noted above, results presented in Figure 3–Figure 5, point to a potential dual role for DMA-2 as both a permissive cue (for secondary and tertiary branches) and instructive signal (for quaternary branches). If this model is correct, then ectopic expression of DMA-2 in other cell types (i.e., non-muscle) should rescue secondary and tertiary branch outgrowth but not the quaternary branch defect in *dma-2* mutants. This experiment should be conducted.

*Reviewer #3:*

*C. elegans* PVD neurons have elaborate dendritic processes that tile the body wall, contacting the hypodermis and body muscles. Prior studies identified 3 cell surface molecules that shapes the elaborate PVD dendritic arbors. DMA-1 functions in PVD neurons while MNR-1 and SAX-7 function in the hypodermis, and all three proteins bind to each other. Here the authors identify a fourth guidance molecule (DMA-2) that is secreted from body muscles, binds directly to the DMA-1/MNR-1/SAX-7 complex, and is also required for patterning PVD dendrites. The phenotypes of mutants lacking MNR-1, SAX-7, and DMA-2 are very similar and additive effects are not observed in double mutants. The authors provide some biochemical data suggesting that DMA-2 strengthens the binding interactions between DMA-1, SAX-7, and MNR-1. Based on these findings, the authors propose that proper PVD dendrite morphogenesis requires DMA-1 receptors to interact simultaneously with 3 distinct ligands (MNR-1/SAX-7/DMA-2), with each ligand controlling distinct aspects of dendrite growth.

The results reported firmly establish DMA-2 as a new guidance molecule for PVD dendrite development. This alone could justify publication in *eLife*; however, I would be more enthusiastic about publication if the author's provided more mechanistic insights into how DMA-2 (and the other guidance molecules) shape dendrite development. The evidence supporting DMA-2's proposed mechanism of action (enhancing formation of the MNR-1/SAX-7/DMA-2 complex) is not fully convincing. How these molecules alter PVD dendrite outgrowth during development (e.g. by time lapse recordings) is not known. These and other concerns are detailed below.

1) The co-IP experiments could be explained by binary interactions between DMA-2/MNR-1/SAX-7/DMA-2. Why do the authors conclude that a complex of all four proteins exists, and that such a complex is required for PVD dendrite development?

2) The effect of DMA-2 on DMA-1 coIP with SAX-7/MNR-1 (Figure 7) should be quantified (% DMA-1 input that is recovered by coIP). The manuscript would be improved if more biochemical experiments were added to convincingly demonstrate a change in binding affinity produced by adding DMA-2.

3) In Figure 7, how are the MNR-1 and DMA-2 bands distinguished (as they both carry the FLAG tag)?

4) This paper focuses on DMA-2's role in promoting PVD "innervation" of body muscles (which I assume means the 4o dendrite branches). Prior papers showed that *dma-1, mnr-1*, and *sax-7* mutants have defects in 2^o^, 3^o^, and 4^o^ dendrite branches. Do *dma-2* mutants also have 2^o^ and 3^o^ branching defects? If this is the case, the authors should modify the text and discussion to describe these results and interpretations more clearly.

5) Do PVD dendrites form specialized contacts with body muscles, as would be expected if they mediate sensation of muscle contraction (and as implied by use of the term "innervation"). The authors should check to see if the Worm Atlas repository of electron micrographs contains useful images of PVD/body muscle contacts.

6) The image shown in Figure 3 suggests that muscle specific dma-2 knockout (using muscle CRE drivers) produces a weaker phenotype than the constitutive *dma-2* null mutant (maybe less 2^o^ and 3^o^ branch defects). Is this the case? Does that imply that DMA-2 expression in other tissues is required for PVD development?

7) Does muscle expression of DMA-2 rescue (Figure 5) rescue 2^o^ and 3^o^ branch defects? If so, the authors should modify the Discussion to indicate that DMA-2 secreted from muscles acts at a distance to promote 2^o^ and 3^o^ branches in PVD dendrites.

8) The distribution of the DMA-2/GFP (Figure 3 S2 C-H) and its rescuing activity (Figure 3—Figure supplement 2B) should be quantified. Is DMA-2/GFP's localization significantly altered in *sax-7* or *mnr-1* mutants?

9) Is expression of the *dma-2* promoter altered in *sax-7* and *mnr-1* mutants? This control is necessary to interpret the experiments shown in Figure 3—figure supplement 2.

10) The authors conclude that DMA-2 is an instructive cue because a transgene expressing DMA-2 in seam cells increases PVD dendrite contact with seam cells (Figure 5). But Figure 3—figure supplement 2 shows that wild type DMA-2/GFP (expressed with its native promoter) stains seam cells. So, why don't wild type PVD neurons innervate seam cells?

11) Does seam cell expressed DMA-2 rescue 2^o^ and 3^o^ branch defects?

12) Could coIP of DMA-1 with the SAX-7/CTF1 fragment (Figure 6) be mediated by an interaction between the FL and CTF1 SAX-7 proteins? If so, one cannot exclude that Nterminal Ig domains are not required for the binding (subsection “DMA-2 physically interacts with SAX-7”).

13) Discussion refers to DMA-1 signaling and DMA-1 "activation" by its ligands. Is it possible that these molecules mediate only adhesive interactions, and may not activate cytoplasmic signaling pathways?

---

## [Author Response]

[…]

*Major Revision:*

*1) The results shown in Figure 3, Figure 4 and Figure 5 suggest that DMA-2 might also function as a permissive signal to promote outgrowth of secondary and tertiary branches. In Figure 3, targeted ablation of DMA-2 expression in muscle (with a muscle-specific cre driver) significantly reduces the number of quaternary branches but secondary and tertiary branches appear largely normal. This result stands in contrast to the dma-2 mutant phenotype which shows highly aberrant secondary and tertiary branch defects in addition to the absence of quaternary branches (Figure 1). Although PVD quaternary branch outgrowth is selectively rescued in a dma-2 mutant for a single DMA-2-expressing body muscle cell (Figure 4), secondary and tertiary branch outgrowth in other regions is also apparently restored. Finally, Figure 5 shows aberrant innervation of lateral seam cells with local over expression of DMA-2 but again, secondary and tertiary branch morphology is also partially rescued in this case despite the lack of DMA-2 expression in muscle. One possibility is that ectopic expression of DMA-2 in other cell types (i.e., non-muscle) may rescue secondary and tertiary branch outgrowth but not the quaternary branch defect in dma-2 mutants. Indeed, it would be straightforward to test the idea that DMA-2 could also function as a permissive cue by expressing it in another cell type (e.g., ventral cord motor neurons). Importantly, but not the only experiment that should be performed, does seam cell expressed DMA-2 rescue 2^o^ and 3^o^ branch defects? These experiments should be conducted.*

We thank the reviewers for the insightful comments. We agree that LECT-2 (named as DMA-2 in the previous version of our manuscript) functions as a long-range guidance cue for dendrite morphogenesis of the 2^o^ and 3^o^ branches, and a short-range cue to instruct the growth of 4^o^ dendrites in vivo. We have performed new experiments including the ones suggested by the reviewers in the revised manuscript to support these conclusions. We want to emphasize that LECT-2 acts as a co-ligand together with SAX-7 and MNR-1 for the DMA-1 receptor in both the “long-range” and “short-range” functions. The distinction between the ranges is likely due to the different extracellular environments where the 2^o^ /3^o^ and 4^o^ dendrites branches form. The 2^o^ /3^o^ branches form against the surface of the body cavity, through which secreted molecules diffuse easily up and down the worm body. In contrast, the 4^o^ branches locate within the narrow space between body wall muscles and skin/epidermal cells. We suspect that endogenous LECT-2 secreted from muscles has very limited mobility in the skin-muscle interface and therefore acts as a short-range cue for the 4^o^ branches. Below are the experiments that we added on this issue.

1) We have added quantification for the number of 2^o^ dendrites and 3^o^ dendrites for the muscle-specific conditional knock-out (cKO) experiments (Figure 5). Compared to the *lect-2* null mutants, muscle cKO animals contained similar number of 3^o^ dendrites, suggesting that 3^o^ dendrite formation does not strictly rely on muscle secreted LECT-2. Interestingly, in the muscle cKO animals, the number of 2^o^ dendrites was significantly increased (Figure 5), However, significant portion of 2^o^ dendrites were short, misguided and did not reach the muscle border. Hence the number of 3^o^ dendrites was not significantly different from wild type in one line and subtly increased in another (Figure 5). These results suggest that the muscle secreted LECT-2 is absolutely required for 4^o^ but not for 2^o^ and 3^o^ branch formation. The muscle secreted LECT-2 might have a function in reducing ectopic 2^o^ branches. One implication of these results is that there might be other cells that express LECT-2 in addition to the body wall muscles. After carefully examining the transcriptional reporter of *lect-2*, we identified a small number of head neurons and ventral nerve cord neurons as *lect-2*-expressing cells (Figure 4 and Figure 4—figure supplement 1). We suspect that the rescue of 2^o^ and 3^o^ dendrites in muscle cKO animals was due to LECT-2 expression from these neurons.

2) To more directly test the idea of local and long-range activity, we utilized the mosaic nature of the muscle promoter-driven transgene in transgenic animals in which only a single or a few muscle cells express *lect-2*. In the original submission, we showed that single cell expression of *lect-2* rescued the 4^o^ branches innervating the same muscles but not adjacent or distant muscles. Following the reviewers’ suggestion, we now quantified the number of 2^o^ and 3^o^ dendrites for the *lect-2*(+) zone and *lect-2(-)* zone. We found that different from the 4^o^ branches, the 3^o^ branches were significantly rescued in both zones (Figure 6). The quantification of the 2^o^ branches in these experiments were confounded by the large number of disorganized 2^o^ branches in the *lect-2* mutants (Figure 6). However, it is clear that the number of 2^o^ branches that gave rise to 3^o^ branches was dramatically increased in both the *lect-2*(+) region and nearby regions. Consistent with the cKO results, these results demonstrate that LECT-2 expressed from a single muscle cell, can function over long distance to form 2^o^ and 3^o^ dendrites but act locally to instruct the innervation of 4^o^ branches to individual muscles.

3) Similarly, seam cell-expressed LECT-2 completely rescued the 3^o^ but not 4^o^ dendrites. We have added quantification for the rescue experiment by expressing LECT-2 in the seam cells (Figure 6—figure supplement 1).

4) We found that over-expressing LECT-2 from multiple tissues, including PVD neurons, epidermal cells (skin), body wall muscles and pharyngeal muscles robustly rescued the dendrite patterning of 2^o^, 3^o^ and 4^o^ dendrites (Figure 6—figure supplement 1). In these experiments, high level of LECT-2 was likely generated from the transgene overexpression, which might lead to sufficient LECT-2 to diffuse into the space between skin and body wall muscle cells and rescue the formation of 4^o^ dendrite. These results also argue that the muscle expressed LECT-2 cannot be the only “instructive cue” to pattern the dendrites because expression of *lect-2* from different tissues (presumably forming different LECT-2 patterns) can rescue the dendrite morphology. We discussed these results in the Discussion (subsection “LECT-2 acts both locally and globally to guide and pattern PVD dendrites”, second paragraph).

*2) Prior papers showed that dma-1, mnr-1, and sax-7 mutants have defects in 2^o^, 3^o^, and 4^o^ dendrite branches. Do dma-2 mutants also have 2^o^ and 3^o^ branching defects? The authors should quantify these effects rigorously and the authors should modify the text and discussion to describe these results and interpretations more clearly. This should be inclusive of the mutant phenotype, rescue and overexpression experiments.*

We thank the reviewers for the suggestion. *lect-2/dma-2* mutants also showed fully penetrant 2^o^ and 3^o^ branching defects. In the original submission, we described the *lect-2* phenotype as “indistinguishable from those of *sax-7* and *mnr-1*”. Our previous papers have shown that *sax-7* and *mnr-1* both have fully penetrant defects in 2^o^ and 3^o^ dendrites. We have added the quantification for the defects of 2^o^ and 3^o^ dendrites (shown in Figure 1 for mutant phenotype, Figure 2 and Figure 6 for rescue and Figure 6—figure supplement 1 for overexpression experiments). We also modified the text and Discussion to describe these results and interpretations.

*3) The distribution of the DMA-2/GFP (Figure 3—figure supplement 2C-H) and its rescuing activity (Figure 3—figure supplement 2B) should be quantified. Is DMA-2/GFP's localization significantly altered in sax-7 or mnr-1 mutants?*

To stringently characterize the expression of *lect-2*, we have generated a yfp knock-in strain to tag the endogenously expressed LECT-2 using CRIPSR/Cas9 method. This knock-in strain generated similar results as the transgene that we used in the original submission. Nevertheless, we replaced the expression data with this knock-in line. The endogenously-expressed YFP::LECT-2 was likely to be largely functional as the animals of the knock-in line showed normal menorah structures with slightly increased 4^o^ branch numbers (Figure 4). The expression pattern was very similar to what we observed using the LECT-2::GFP high-copy transgenes (Figure 4 and Figure 8). In addition to the coelomocyte staining, YFP::LECT-2 labeled stripe patterns, which was completely lost in the *sax-7* mutant, but not changed in *mnr-1* mutants (Figure 8).

*4) Many results were not quantified, and sample sizes were only summarized (e.g. "at least 12 animals") for those that were quantified. Please quantify all reported results and please indicate specific sample sizes for every genotype in each figure.*

We have added quantification for most of the results we reported in the revised manuscript. We also indicated specific sample sizes for every genotype in each figure.

*5) The coIP data are nice but do not document a single complex of all 4 proteins. The coIPs also do not convincingly show that DMA-2 alters the affinity of the complex for the other proteins. More specifically, there is a failure to replicate a previous result showing that the receptor complex can form in the absence of DMA-2. The results shown here suggest that DMA-2 is required (and in that sense clearly "enhances" interaction). It should be possible to show a quantitative dependence on DMA-2. Beyond this, the interpretation of the biochemistry should be more circumspect.*

We thank the reviewers to point these out. To test whether LECT-2/DMA-2 forms a four-protein complex with DMA-1, SAX-7 and MNR-1 in vivo, we collaborated with Dr. Kevin Xiang’s group and performed quantitative, single molecule pull-down experiments using worm extracts. In these experiments, endogenous DMA-1 was tagged with FLAG using CRISPR/Cas9 method. LECT-2 and SAX-7 were tagged with GFP and mCherry, respectively. Our new results clearly showed that DMA-1, LECT-2 and SAX-7 form a multi-protein complex. Due to technical difficulty, we were not able to also label MNR-1 with a third fluorescent tag. However, we found that the formation of DMA-1-LECT-2-SAX-7 complex was completely MNR-1-dependent, which strongly suggests that MNR-1 is also an essential component of the multi-protein complex (Figure 9).

To quantitatively test whether inclusion of LECT-2 enhances the binding efficiency between DMA-1, SAX-7 and MNR-1, we performed single molecule pull-down experiments using S2 cell lysates with or without adding LECT-2. We found that DMA-1 interacted with SAX-7 and MNR-1, even in the absence of LECT-2, confirming the published results. Importantly, adding LECT-2 dramatically enhances the binding efficiency between DMA-1, SAX-7 and MNR-1 by roughly 40 fold (Figure 10).

*6) The Discussion says: "It is interesting to note that the three ligands from two tissues contain distinct information for PVD dendrite patterning." If the mutants have identical phenotypes, this can't be true or, at a minimum, is very confusing to the reader. The authors should address this issue experimentally or through modification of the text.*

We apologize for the confusion. We have now modified the text.